# A Regret-Variance Trade-Off in Online Learning

**Dirk van der Hoeven**
dirk@dirkvanderhoeven.com
Dept. of Computer Science
Università degli Studi di Milano, Italy

**Nikita Zhivotovskiy**
zhivotovskiy@berkeley.edu
Dept. of Statistics
University of California, Berkeley

**Nicolò Cesa-Bianchi**
nicolo.cesa-bianchi@unimi.it
Dept. of Computer Science
Università degli Studi di Milano, Italy

## Abstract

We consider prediction with expert advice for strongly convex and bounded losses, and investigate trade-offs between regret and "variance" (i.e., squared difference of learner's predictions and best expert predictions). With $K$ experts, the Exponentially Weighted Average (EWA) algorithm is known to achieve $O(\log K)$ regret. We prove that a variant of EWA either achieves a *negative* regret (i.e., the algorithm outperforms the best expert), or guarantees a $O(\log K)$ bound on *both* variance and regret. Building on this result, we show several examples of how variance of predictions can be exploited in learning. In the online to batch analysis, we show that a large empirical variance allows to stop the online to batch conversion early and outperform the risk of the best predictor in the class. We also recover the optimal rate of model selection aggregation when we do not consider early stopping. In online prediction with corrupted losses, we show that the effect of corruption on the regret can be compensated by a large variance. In online selective sampling, we design an algorithm that samples less when the variance is large, while guaranteeing the optimal regret bound in expectation. In online learning with abstention, we use a similar term as the variance to derive the first high-probability $O(\log K)$ regret bound in this setting. Finally, we extend our results to the setting of online linear regression.

## 1 Introduction

In the online learning protocol, the learner interacts with an unknown environment in a sequence of rounds. In each round $t = 1, 2, \ldots, T$ the learner makes a prediction $\widehat{y}_t \in [-\frac{1}{2}M, \frac{1}{2}M]$ and suffers loss $\ell_t(\widehat{y}_t)$, where $\ell_1, \ell_2 \ldots$ is a sequence of differentiable loss functions unknown to the learner. The goal is to achieve small regret $\mathcal{R}_T$, defined by $\mathcal{R}_T = \sum_{t=1}^{T} \left( \ell_t(\widehat{y}_t) - \ell_t(y_t^\star) \right)$ where $y_t^\star$ are the predictions of some reference forecaster. We consider two special cases of online learning. In prediction with expert advice, the learner receives the predictions $y_t(1), \ldots, y_K(t) \in [-\frac{1}{2}M, \frac{1}{2}M]$ of $K$ experts at the beginning of each round $t$. The learner then predicts with a convex combination $\widehat{y}_t = \sum_{i=1}^{K} p_t(i) y_t(i)$ of the experts' predictions, where $\boldsymbol{p}_t = \left( p_t(1), \ldots, p_t(K) \right)$ is a probability distribution over experts updated after each round. The goal in this setting is to have small regret with respect to the predictions $y_t^\star = y_t(i^\star)$ of any fixed expert $i^\star$. In online linear regression, the learner has access to a feature vector $\boldsymbol{x}_t \in \mathbb{R}^d$ at each round. This is used to compute predictions $\widehat{y}_t = \langle \boldsymbol{w}_t, \boldsymbol{x}_t \rangle$, where $\boldsymbol{w}_t \in \mathbb{R}^d$ is a parameter vector to be updated at the end of the round. The goal

is to have small regret with respect to the predictions $y_t^\star = \langle \boldsymbol{u}, \boldsymbol{x}_t \rangle$ of any fixed linear forecaster $\boldsymbol{u} \in \mathbb{R}^d$ such that $\|\boldsymbol{u}\|_2 \leq D$ and $|y_t^\star| \leq \frac{1}{2}M$ for all $t$.

In both settings, we assume that the losses $\ell_t$ are $\mu$-strongly convex, which is to say that there exists $\mu > 0$ such that for all $y, x \in [-\frac{1}{2}M, \frac{1}{2}M]$,

$$\ell_t(y) - \ell_t(x) \leq (y - x)\ell_t'(y) - \frac{\mu}{2}(x - y)^2 \qquad t = 1, \ldots, T , \tag{1}$$

where we write $\ell_t'$ to denote the derivative of $\ell_t$. Without loss of generality, throughout the paper we assume that $\mu \leq 2$.

A typical example of a strongly convex loss is the squared loss $\ell_t(y) = (y - y_t)^2$ with $\mu = 2$. While standard online learning algorithms, such as, for example, Exponentially Weighted Average (EWA) [Vovk, 1990, Littlestone and Warmuth, 1994], are directly applied to the (mixable) losses $\ell_t$, we take a different approach. Our algorithms provide bounds on the linearized regret $\widetilde{\mathcal{R}}_T$ of the form

$$\widetilde{\mathcal{R}}_T = \sum_{t=1}^T (\widehat{y}_t - y_t^\star)\ell_t'(\widehat{y}_t) \leq \frac{C_T}{\eta} + B_T + \eta \sum_{t=1}^T (\widehat{y}_t - y_t^\star)^2. \tag{2}$$

for some $\eta \in (0, H]$ to be chosen freely by the learner and where $H, C_T, B_T \geq 0$ are problem-specific parameters. In the expert setting, several algorithms provide such a guarantee: Squint [Koolen and Van Erven, 2015], Adapt-ML-Prod [Gaillard et al., 2014], BOA [Wintenberger, 2017], Squint+C and Squint+L [Mhammedi et al., 2019]. In online linear regression, the MetaGrad algorithm [Van Erven et al., 2021] and its variants by Mhammedi et al. [2019], Wang et al. [2020], Chen et al. [2021] satisfy bounds like (2). Although in both settings the aforementioned algorithms achieve the optimal tuning of $\eta$ in (2) without any preliminary information, our applications do not require this tuning property. As a consequence, the algorithms we derive in Section 3 are simpler and have slightly better guarantees. More importantly, unlike the aforementioned algorithms, we focus on exploiting the curvature of the loss. In particular, by combining equations (1) and (2), we obtain the following lemma, which features the central inequality of our work.

**Lemma 1.** *Consider a sequence $\ell_1, \ldots, \ell_T$ of $\mu$-strongly convex differentiable losses. Suppose that predictions $\widehat{y}_t$ guarantee the second-order bound* (2). *Then*

$$\mathcal{R}_T \leq \frac{C_T}{\eta} + B_T - \left(\frac{\mu}{2} - \eta\right) \sum_{t=1}^T (\widehat{y}_t - y_t^\star)^2.$$

Next, we show a prototypical example of the bounds we derive in the rest of this work.

**Example 1.** *Consider the expert setting with the squared loss, $\ell_t(y) = (y - y_t)^2$, where $y, y_t \in [-1, 1]$ for all $t \geq 1$. Then the predictions $\widehat{y}_t$ of our algorithm satisfy*

$$\mathcal{R}_T \leq 32 \log(K) - \frac{1}{2} \sum_{t=1}^T (\widehat{y}_t - y_t^\star)^2 . \tag{3}$$

Even though (3) seems relatively inconsequential, we exploit the negative variance term in several applications. A straightforward implication is that (3) recovers the usual *constant* regret bound $\mathcal{R}_T = O(\log(K))$. More interestingly, however: if we attain the worst-case performance $O(\log(K))$, then the variance is small and bounded by $O(\log(K))$, if the variance is large enough, then the regret becomes negative. For the role of negative variance terms in the analysis of statistical and online learning, we refer to Section 2.

**Remark.** *Negative quadratic terms similar to the one in Lemma 1 appear in online convex optimization, for example in the analysis of the online gradient descent [Hazan, 2016, Theorem 3.3]. However, as far as we are aware, in online convex optimization the algorithms are not tuned to obtain a negative term in the regret bounds. In this paper we argue that for online regression, and especially for prediction with expert advice, setting $\eta < \mu/2$ in Lemma 1 and thus obtaining a negative term can be a powerful tool, which plays a central role in all of our applications. In prediction with expert advice the standard approach to exploit the curvature of the loss is through mixability rather than through the negative quadratic terms. In particular, we show that the regret bound in (3) cannot be achieved by EWA with a fixed learning rate, despite the fact that constant regret is achievable by this algorithm via mixability.*

**Proposition 1** (Informal). *Consider the setup of online prediction with expert advice and bounded strongly convex losses. The bound* (3) *cannot be achieved by the standard EWA algorithm.*

For the sake of presentation, we defer the formal description of this result to Section 3.1. The proof of Proposition 1 is motivated by a result of Audibert [2007] on the sub-optimality of online to batch converted EWA in deviation.

**Contributions and Outline.** In Section 4, we show our first three applications of Lemma 1 in the framework of statistical learning. In the expert setting, we show that online to batch conversion may be stopped early if the empirical variance of our predictions is sufficiently large. In particular, in high-variance regimes we may stop early because the excess risk bound is negative with high probability, and when the variance is small, we recover the optimal excess risk bound up to a $\log \log T$ factor. By exploiting the negative variance term again, we show an optimal high-probability excess risk bound for online to batch conversion of algorithms that satisfy (2). The optimal high-probability excess risk bound (called the optimal rate of model selection aggregation) was previously known to be achieved by several estimators appearing in [Audibert, 2007, Lecué and Mendelson, 2009, Lecué and Rigollet, 2014] whose analyses are specific to the statistical learning setup. Our result can also be seen as a simplification and strengthening of a result by Wintenberger [2017]. We also show a high-probability excess risk bound for online to batch conversion of an online regression algorithm in the bounded setup, which is to say that both the feature vectors and derivatives of the losses are bounded. It was previously shown by Mourtada et al. [2021] that online to batch converted versions of the optimal Vovk-Azoury-Warmuth forecaster [Vovk, 2001, Azoury and Warmuth, 2001] have constant excess risk with constant probability.

In Section 5, we use the negative variance term to counteract corrupted feedback in online learning. Our result complements previous results where losses are assumed to be stochastic and corrupted by an adversary—see, e.g., Lykouris et al. [2018], Zimmert and Seldin [2021], Amir et al. [2020], Ito [2021]. In Section 6 we consider the selective sampling setting [Atlas et al., 1989], where the learner's goal is to control regret while saving on the number of time the current loss is observed. We show that the optimal bound can be recovered while only observing a fraction of all losses if losses are observed proportionally to the cumulative variance. Finally, in Section 7 we discuss how our ideas are not limited to online learning with strongly convex losses, but may also be applied to online learning with abstention and to online multiclass classification. All bounds in the main text with suppressed constants have detailed statements in the appendix. Before discussing our applications, we introduce some notation and discuss the related work. The algorithms that we use to derive most of our results are introduced in Section 3.

**Notation.** We use the standard $O(\cdot)$ to suppress constants and use $\widetilde{O}(\cdot)$ to suppress constants and $\log(T)$-factors. We use $\log(\cdot)$ to denote the logarithm with base $e$. The symbol $\mathbb{1}[A]$ denotes the indicator of the event $A$. For an integer $K$ we denote $[K] = \{1, \ldots, K\}$. We use $p(i) \propto g(i)$ to denote $p(i) = g(i) \Big/ \left( \sum_{i'=1}^{K} g(i') \right)$ and $\partial_y \ell(y, Y)$ to denote the derivative of $\ell$ with respect to $y$. The symbol $I$ denotes the identity matrix whose dimensions are clear from the context. For a set of random variables $Y, X_1, \ldots, X_{t-1}$ we write $\mathbb{E}_{t-1}[Y] = \mathbb{E}[Y|X_1, \ldots, X_{t-1}]$.

## 2 Related work

**Online learning with losses with curvature.** For a thorough introduction to online learning, we refer the reader to [Cesa-Bianchi and Lugosi, 2006, Hazan, 2016, Orabona, 2019]. Strongly convex losses are a special case of mixable losses. With mixable losses, EWA on a finite set of experts achieves $O(\log(K))$ regret—see, e.g., [Cesa-Bianchi and Lugosi, 2006, Mhammedi and Williamson, 2018]. Mhammedi and Williamson [2018] observe that—in some cases—regret may be negative for mixable losses, but they do not explore this topic further.

**The role of the negative terms due to curvature in online and statistical learning.** Although bounds of the form (3) are not explicitly present in the literature, the appearance of negative terms is common when proving fast rates in statistical learning with squared loss. In particular, the *empirical star* algorithm of Audibert [2007]—as well as other aggregation algorithms [Lecué and Mendelson, 2009, Lecué and Rigollet, 2014, Wintenberger, 2017]—exploit the curvature of the loss

---

**Algorithm 1:** An algorithm for prediction with expert advice

---

**Input** $\eta \in (0, \frac{1}{2}]$, $M > 0$ ;
**Initialize** $\gamma = \frac{\eta}{M^2}$, $p_1(i) = \frac{1}{K}$ for all $i$ ;
**for** $t = 1, \ldots, T$ **do**
    Receive expert predictions $y_t(1), \ldots, y_t(K)$
    Predict $\widehat{y}_t = \sum_{i=1}^{K} p_t(i) y_t(i)$
    Receive $g_t$ and $\kappa_t$
    Set $\widetilde{\ell}_t(i) = \gamma(y_t(i) - \widehat{y}_t)g_t + \kappa_{t-1}(\gamma(y_t(i) - \widehat{y}_t)g_t)^2$
    Set $p_{t+1}(i) \propto \exp(-\kappa_t \sum_{s=1}^{t} \widetilde{\ell}_s(i))$

---

through the negative term which compensates the variance term. We also refer to some more recent statistical learning results whose analysis is based on the same idea [Mendelson, 2019, Bousquet and Zhivotovskiy, 2021, Puchkin and Zhivotovskiy, 2022, Saad and Blanchard, 2021] . Similarly, in the context of online learning the negative quadratic term appears in Rakhlin and Sridharan [2014], where the so-called *sequential offset Rademacher complexity* is studied. Van Erven et al. [2021] also obtain a bound that is very similar to the bound in Lemma 1 (specifically in the proof of their Theorem 1), but they choose $\eta$ to match the negative term in their bound. Importantly, in these papers the role of the negative term is only to get the fast rate by compensating the variance term. In contrast, in our case the negative variance terms also appear in the final regret bound and play their role in applications.

**Suboptimality of EWA for prediction with expert advice.** In the setup of prediction with expert advice, the classical Exponentially Weighted Average (EWA) algorithm [Vovk, 1990, Littlestone and Warmuth, 1994] is known to give a constant regret in the case of strongly convex losses. However, despite being optimal in this setting, this algorithm has several known drawbacks. In the case of general losses, EWA does not deliver the second-order bound (2). As a result, unlike more advanced algorithms such as Squint [Koolen and Van Erven, 2015], EWA with a fixed learning rate (and even a decreasing learning rate) cannot adapt to certain benign stochastic environments where, for example, the Bernstein assumption holds [Mourtada and Gaïffas, 2019]. The second source of suboptimality comes from online to batch conversions in the strongly convex case. Although in the statistical setting EWA performs optimally in expectation, it does not do so with high probability [Audibert, 2007]. As a matter of fact, it will be clear from our analysis that this is related to the fact that EWA does not satisfy a bound of the form (3) (see Theorem 1). This problem is one of the motivations behind the work of Wintenberger [2017].

**Exploiting negative regret.** The possibility of getting a negative excess risk has been recently explicitly exploited by Puchkin and Zhivotovskiy [2022, see their Algorithm 3.3 and Theorem 3.4] in the setup of active learning with abstentions. In the context of online to batch conversion of online learning algorithms, a similar idea is exploited in Section 4.1. Moreover, our selective sampling results in Section 6 are of the same flavor.

## 3 Our Algorithms

In the following, $g_1, g_2, \ldots$ are real numbers in a bounded interval. However, in most applications we use $g_t = \ell'_t(\widehat{y}_t)$. The algorithms in this section are simplified versions of algorithms in the literature. Namely, in Section 3.1 we present a simplified version of Squint [Koolen and Van Erven, 2015] and in Section 3.2 we present a simplified version of MetaGrad [Van Erven et al., 2021]. The simplifications lie in the tuning of the learning rate. Squint and MetaGrad optimize the learning rate online at a small cost, whereas in our applications we only need a fixed learning rate that is known in advance, in which case we do not pay the small cost for optimizing the learning rate. The proofs of the results in this section are postponed to Appendix A.

**Algorithm 2:** An algorithm for online linear regression

---

**Input** $\eta > 0$, $\sigma > 0$, $G > 0$, $Z > 0$ ;
**Initialize** $\gamma = \frac{\eta}{G^2}$, $\boldsymbol{w}_1 = \boldsymbol{0}$, and $\Sigma_1^{-1} = \frac{1}{\sigma} I$ ;
**for** $t = 1, \ldots, T$ **do**
    Receive $\boldsymbol{x}_t$
    Set $\mathcal{W}_t = \bigcap_{s=1}^t \{\boldsymbol{w} : |\langle \boldsymbol{w}, \boldsymbol{x}_s \rangle| \leq Z\}$
    Set $\boldsymbol{w}_t = \mathrm{argmin}_{\boldsymbol{w} \in \mathcal{W}_t} (\boldsymbol{w} - \widetilde{\boldsymbol{w}}_t)^\top \Sigma_t^{-1} (\boldsymbol{w} - \widetilde{\boldsymbol{w}}_t)$
    Predict $\widehat{y}_t = \langle \boldsymbol{w}_t, \boldsymbol{x}_t \rangle$
    Receive $g_t$ and $\kappa_t$
    Set $\boldsymbol{z}_t = \gamma \boldsymbol{x}_t g_t$
    Set $\Sigma_{t+1}^{-1} = \kappa_t 2 \boldsymbol{z}_t \boldsymbol{z}_t^\top + \Sigma_t^{-1}$
    Set $\widetilde{\boldsymbol{w}}_{t+1} = \boldsymbol{w}_t - \boldsymbol{z}_t \Sigma_{t+1}$

---

### 3.1 A Simple Algorithm for Prediction with Expert Advice

Algorithm 1 is a simplified version of Squint [Koolen and Van Erven, 2015]. The $\kappa_t$ parameter is relevant only to the selective sampling setting, where it is used to control the range of loss estimates. In all other settings we set $\kappa_t = 1$ for all $t$. Note that in round $t$ both $\kappa_t$ and $\kappa_{t-1}$ are used to update the algorithm. This is due to a technicality in the analysis of the algorithm, where a $\kappa_{t-1} \widetilde{\ell}_t(i)$ term appears and we want to use the inequality $x - x^2 \leq \log(1 + x)$ for $|x| \leq \frac{1}{2}$ (specifically in equation (7)). The regret bound of Algorithm 1 can be found in Lemma 2.

**Lemma 2.** *For all $g_1, \ldots, g_T \in [-M, M]$ and $\kappa_0 = \kappa_1 \geq \kappa_2 \cdots \geq \kappa_T$ such that $\kappa_t \in (0, 1]$, the predictions $\widehat{y}_t$ of Algorithm 1 run with input $M$ and $\eta \in (0, \frac{1}{2}]$ satisfy*

$$\sum_{t=1}^T (\widehat{y}_t - y_t(i)) g_t \leq \frac{M^2 \log(K)}{\kappa_T \eta} + \eta \sum_{t=1}^T \kappa_{t-1} (\widehat{y}_t - y_t^\star)^2$$

*provided $\max_i |y_t(i) - y_t^\star| \leq M$ for all $t \geq 1$.*

As an immediate corollary of Lemma 2 and Lemma 1 we have the following regret bound.

**Corollary 1.** *Fix an arbitrary sequence $\ell_1, \ldots, \ell_T$ of $\mu$-strongly convex differentiable losses such that $\max_t |\ell_t'| \leq M$. Provided that $\max_i |y_t^\star - y_t(i)| \leq M$ for all $t \geq 1$, the predictions $\widehat{y}_t$ of Algorithm 1 run with inputs $M$ and $\eta \in [0, \frac{1}{2})$, $\kappa_t = 1$ for all $t$, and feedback $g_t = \ell_t'(\widehat{y}_t)$, satisfy*

$$\mathcal{R}_T \leq \frac{M^2 \log(K)}{\eta} - \left(\frac{\mu}{2} - \eta\right) \sum_{t=1}^T (\widehat{y}_t - y_t^\star)^2 .$$

As an example, let us consider the squared loss, which is 2-strongly convex. In the setup of Example 1, since $|\ell_t'(\widehat{y}_t)| = 2|(\widehat{y}_t - y_t)| \leq 4$, Algorithm 1 with $g_t = \ell_t'(\widehat{y}_t)$ and $\eta = \frac{1}{2}$ gives us the regret bound claimed in Example 1, namely $\mathcal{R}_T \leq 32 \log(K) - \frac{1}{2} \sum_{t=1}^T (\widehat{y}_t - y_t^\star)^2$. Our next result is a formal version of Proposition 1 saying that the above regret bound cannot be achieved by the standard EWA algorithm. For standard notation and explicit details on this algorithm we refer to Appendix E.

**Theorem 1.** *Consider the squared loss and two experts $y_t(1) = 0$ and $y_t(2) = 1$ for all $t \geq 1$. Let $\widehat{y}_t^{EWA}$ be the EWA predictions. There is a sequence $y_1, y_2, \ldots$ such that $y_t \in [0, 1]$, $t \geq 1$ and, for large enough $T$, the regret of EWA with $\eta = \frac{1}{2}$ satisfies $-12 \log T \leq \mathcal{R}_T \leq 2 \log 2$ and, at the same time,*

$$\sum_{t=1}^T (\widehat{y}_t^{EWA} - y_t^\star)^2 \geq T/2 .$$

### 3.2 A Simple Algorithm for Online Regression

In the following we use $y_t(\boldsymbol{u}) = \langle \boldsymbol{u}, \boldsymbol{x}_t \rangle$. We prove a regret bound for Algorithm 2, which is a simplified version of MetaGrad [Van Erven et al., 2021]. The role of the parameter $Z$ in the algorithm is to ensure the predictions $\widehat{y}_t$ are bounded, which will be important in the statistical learning setting in Section 4. Similarly to Algorithm 1, the $\kappa_t$ parameter is only used in the selective sampling setting.

**Lemma 3.** *For all $g_1, \ldots, g_T \in \mathbb{R}$ and $\kappa_1 \geq \cdots \geq \kappa_T \in (0, 1]$, the predictions $\widehat{y}_t$ of Algorithm 2 run with inputs $\eta > 0$, $\sigma = D^2$, $G \geq \max_t |g_t|$, and $Z > 0$*

$$\sum_{t=1}^{T}(\widehat{y}_t - y_t(\boldsymbol{u}))g_t \leq \frac{dG^2}{2\kappa_T\eta} \log\left(1 + D^2\eta^2\left(\max_{t=1,\ldots,T}\|\boldsymbol{x}_t\|_2^2\right)\frac{T}{d}\right) + \frac{G^2}{2\eta} + \eta\sum_{t=1}^{T}\kappa_t(\widehat{y}_t - y_t(\boldsymbol{u}))^2,$$

*for any $\boldsymbol{x}_1, \ldots, \boldsymbol{x}_T \in \mathbb{R}^d$, and for any $\boldsymbol{u} \in \mathcal{W}_T \equiv \bigcap_{t=1}^{T}\{\boldsymbol{w} : |\langle \boldsymbol{w}, \boldsymbol{x}_t\rangle| \leq Z\}$ such that $\|\boldsymbol{u}\|_2 \leq D$.*

**Example 2.** *Consider the setup of Example 1 and suppose that $\max_t \|\boldsymbol{x}_t\|_2, \|\boldsymbol{u}\|_2 \leq 1$ and $M = 1$. We have that $|\ell_t'(\widehat{y}_t)| = |2(\widehat{y}_t - y_t)| \leq 4$ and thus, by Lemma 3, an appropriately tuned Algorithm 2 with $g_t = \ell_t'(\widehat{y}_t)$ satisfies (2) with $C_T = 8 + 8d\log\left(1 + \eta^2\frac{T}{d}\right)$ and $B_T = 0$. Thus, by Lemma 1, setting $\eta = \frac{1}{2}$ gives us*

$$R_T \leq 16 + 16d\log\left(1 + \tfrac{T}{4d}\right) - \tfrac{1}{2}\sum_{t=1}^{T}(\widehat{y}_t - y_t(\boldsymbol{u}))^2.$$

## 4  Statistical Learning

We discuss an application of our general results in the context of statistical learning where we are interested in the generalization of estimators to unseen samples. A tool often used in converting online learning algorithms to the statistical learning setting is online to batch conversion [Cesa-Bianchi et al., 2004]. Let us recall the setup.

Assume that we are given a family $\mathcal{F}$ of real-valued functions defined on the instance space $\mathcal{X}$. We observe $T$ i.i.d. observations $(X_t, Y_t)_{t=1}^T$ distributed according to some unknown distribution $\mathbb{P}$ on $\mathcal{X} \times \mathbb{R}$. Given the loss function $\ell : \mathbb{R}^2 \to \mathbb{R}$, define the *risk $R(f)$* of $f : \mathcal{X} \to \mathbb{R}$ as $R(f) = \mathbb{E}\,\ell(f(X), Y)$, where the expectation is taken with respect to the joint distribution of $X$ and $Y$. We are interested in bounding the *excess risk* $R(\widehat{f}) - \inf_{f \in \mathcal{F}} R(f)$, where $\widehat{f}$ is constructed based on the sample $(X_t, Y_t)_{t=1}^T$. Assume that there is a sequence of predictors $\widehat{f}_1, \ldots, \widehat{f}_T$ trained in an online manner using $(X_t, Y_t)_{t=1}^T$ (that is, $\widehat{f}_k$ depends on $(X_t, Y_t)_{t=1}^{k-1}$) such that almost surely $\sum_{t=1}^{T}\left(\ell(\widehat{f}_t(X_t), Y_t) - \ell(f^\star(X_t), Y_t)\right) \leq R_T$, where $R_T$ is non-random. In this case, a standard online to batch conversion approach gives an in-expectation excess risk bound $\mathbb{E}\,R\left(\frac{1}{T}\sum_{t=1}^{T}f_t\right) - \inf_{f \in \mathcal{F}} R(f) \leq \frac{R_T}{T}$, for any loss convex in its first argument and where the expectation is taken with respect to the learning sample $(X_t, Y_t)_{t=1}^T$. However, getting a high-probability version of this result is a known challenge if one wants to get the fast rate $O\left(\frac{1}{T}\right)$. A standard way of proving a high-probability result is to apply Freedman's inequality for martingales [Kakade and Tewari, 2008] that leads to a variance term scaling as $O\left(\frac{1}{\sqrt{T}}\right)$. For example, Audibert [2007] showed that this is the case if one wants to prove a high-probability excess risk bounds based on EWA. A way to handle the variance term in Freedman's inequality is by exploiting the Bernstein assumption as in Kakade and Tewari [2008]. Unfortunately, this assumption is not necessarily satisfied by the stochastic environments we are considering. The main idea in this section is to use the negative term from Lemma 1 to cancel out this variance term appearing due to Freedman's inequality[1]. We use the following notation when applying the online algorithms in the statistical setting: $\ell_t(\cdot) = \ell(\cdot, Y_t)$ and $y_t(f) = f(X_t)$.

### 4.1  Statistical Learning: Model Selection Aggregation

In this section, we discuss the application of our results to the model selection (MS) aggregation. This setup was introduced by Nemirovski [2000] and further studied by Tsybakov [2003] and by Audibert [2007], Lecué and Mendelson [2009], Lecué and Rigollet [2014], Wintenberger [2017], Mourtada et al. [2021] among other works. We introduce some notation. Assume that we are given a finite dictionary $\mathcal{F} = \{f_1, \ldots, f_K\}$ of real-valued functions defined on the instance space $\mathcal{X}$. We observe $T$ i.i.d. observations $(X_t, Y_t)_{t=1}^T$ distributed according to some unknown distribution $\mathbb{P}$ on $\mathcal{X} \times \mathbb{R}$.

---

[1]We remark that Wintenberger [2017] uses a similar but technically more involved idea to compensate the variance of predictions using the term appearing because of the curvature of the loss.

---

**Algorithm 3:** Early Stopping online to batch for Model Selection Aggregation

---

**Input** $T$, $M$, $\eta$, stopping threshold $\mathcal{S}$ ;
**Initialize** $S = 0$, provide $\eta$, and $M$ as input for Algorithm 1 ;
**while** $S < T$ *and* $\mathcal{S} > \frac{\mu}{8} \min_{f \in \mathcal{F}} \sum_{t=1}^{S} (\widehat{y}_t - f(X_t))^2$ **do**
    Receive $X_t$ and send $f_1(X_t), \ldots, f_K(X_t)$ as expert predictions to Algorithm 1
    Receive $\boldsymbol{p}_t$ and $\widehat{y}_t = \sum_{i=1}^{K} p_t(i) f_i(X_t)$ from Algorithm 1
    Predict $\widehat{y}_t$ and receive $\ell_t$
    Send $g_t = \ell_t'(\widehat{y}_t)$ and $\kappa_t = 1$ to Algorithm 1
    Set $S = S + 1$
**Output** $\widehat{f} = \frac{1}{S} \sum_{t=1}^{S} \sum_{i=1}^{K} p_t(i) f_i$ ;

---

Given the loss function $\ell : \mathbb{R}^2 \to \mathbb{R}$, define the *risk* $R(f)$ of $f : \mathcal{X} \to \mathbb{R}$ as $R(f) = \mathbb{E}\,\ell(f(X), Y)$, where the expectation is taken with respect to the joint distribution of $X$ and $Y$. In the model selection aggregation, one is interested in constructing an estimator $\widehat{f}$ based on the random sample $(X_t, Y_t)_{t=1}^{T}$ such that, with probability at least $1 - \delta$,

$$R(\widehat{f}) - \min_{f \in \mathcal{F}} R(f) = O\left(\frac{\log(K) + \log(1/\delta)}{T}\right) , \tag{4}$$

under appropriate boundedness and curvature assumptions on the loss function $\ell$. Analogously to Tsybakov [2003], the bound of the form (4) will be called the *optimal rate of aggregation*. We make use of a variant of online to batch conversion [Cesa-Bianchi et al., 2004] where we stop the procedure early if the empirical variance of predictions is sufficiently large. We sketch the idea. Let $S$ be the number of samples we have used before we terminated the procedure. We use Algorithm 1 as our aggregation procedure and use $\widehat{f} = \frac{1}{S} \sum_{t=1}^{S} \sum_{i=1}^{K} p_t(i) f_i$. By Jensen's inequality we have $R(\widehat{f}) \leq \frac{1}{S} \sum_{t=1}^{S} \mathbb{E}_{t-1}\left[\ell_t\left(\sum_{i=1}^{K} p_t(i) f_i(X_t)\right)\right]$. To motivate stopping early, observe that if the empirical variance in Lemma 1 is sufficiently large, we may conclude that the excess risk is negative and we have outperformed the best $f \in \mathcal{F}$. The result can be found in Theorem 2 below, whose proof is implied by Theorem 8 in Appendix B.

**Theorem 2.** *Suppose that for all $f \in \mathcal{F}$ $|f(X)| \leq \frac{1}{2}$ almost surely, that $|\partial_y \ell(y, Y)| \leq 1$ almost surely for all $y$ such that $|y| \leq \frac{1}{2}$, and that $\ell$ is $\mu$-strongly convex in its first argument. Then, with probability at least $1 - \delta$, Algorithm 3 with input parameters $T$, $\mathcal{S} = O\left(\frac{\log(K) + \log(\log(T)/\delta))}{\mu}\right)$, $\eta = \frac{\mu}{8}$, and $M = 1$ satisfies*

$$R(\widehat{f}) \leq \begin{cases} \min_{f \in \mathcal{F}} R(f) & \text{if } S < T \\ \min_{f \in \mathcal{F}} R(f) + O\left(\frac{\log(K) + \log(\log(T)/\delta))}{\mu T}\right) & \text{if } S = T, \end{cases}$$

*where $S$ is the number of steps of Algorithm 3.*

When Algorithm 3 terminates at step $S = T$, we recover the optimal high probability bound for model selection aggregation (4) up to an additive $\log \log T$ term. However, when $S = T$ our bound tells us slightly more, because we know that for all $t' < T$, $\min_{f \in \mathcal{F}} \sum_{t=1}^{t'} (\widehat{y}_t - f(X_t))^2 = O(\log K + \log \log T)$, which means that on the sequence $(X_t, Y_t)_{t=1}^{T}$ our predictions $\widehat{y}_t$ are essentially following the prediction of the currently best expert at each round.

In the special case where we are solely interested in the best possible performance of the online to batch conversion of Algorithm 1, we can remove the $\log \log T$ term appearing in the previous bound. We remark that, apart from the work of Wintenberger [2017], no known analysis based on the online to batch conversion achieved the optimal rate of aggregation (4). We also believe that our analysis is simpler than for previously known algorithms. The result can be found in Theorem 3 below, whose result is implied by Theorem 9 in Appendix B.

**Theorem 3.** *Suppose that for all $f \in \mathcal{F}$ $|f(X)| \leq \frac{1}{2}$ almost surely, that $|\partial_y \ell(y, Y)| \leq 1$ almost surely for all $y$ such that $|y| \leq \frac{1}{2}$, and that $\ell$ is $\mu$-strongly convex in its first argument. Then, with probability at least $1 - \delta$, Algorithm 3 with input parameters $T$, $\eta = \frac{\mu}{4}$, $\mathcal{S} = \infty$, and $M = 1$,*

*guarantees*

$$R(\widehat{f}) - \min_{f \in \mathcal{F}} R(f) = O\left(\frac{\log(K) + \log(1/\delta)}{\mu T}\right).$$

## 4.2 Statistical Learning: Regression

We consider the statistical learning setting where one has access to $T$ i.i.d. samples of pairs $(X_t, Y_t) \in \mathbb{R}^d \times \mathbb{R}$. In this case, we consider $\mathcal{F} \subseteq \{x \mapsto \langle w, x \rangle : w \in \mathbb{R}^d\}$. For $w \in \mathbb{R}^d$ we define the risk as $R(w) = \mathbb{E}\left[\ell(\langle w, X \rangle, Y)\right]$. We assume that $\ell$ is $\mu$-strongly convex in its first argument.

To the best of our knowledge there are no known high probability excess risk bounds in regression based on online to batch conversions with convergence rate $O\left(\frac{d \log(T)}{T}\right)$. We provide such a result in the bounded setup where the feature vectors, derivatives of the losses, and the norm of the reference vector are bounded. Similarly to before, for a result that holds with high probability, one needs to control the cumulative variance of our prediction. For standard online learning algorithms the control of the variance may prove troublesome. For example, Mourtada et al. [2021] showed that a version of Vovk-Azoury-Warmuth forecaster [Vovk, 2001, Azoury and Warmuth, 2001] may have a $O(1)$ excess risk bound with constant probability, whereas in expectation the Vovk-Azoury-Warmuth forecaster guarantees a $O\left(\frac{d \log(T)}{T}\right)$ excess risk bound. Instead, we leverage the negative empirical variance of Lemma 1 to control the variance of the online to batch conversion, leading to the following excess risk bound, whose result is implied by Theorem 10 in Appendix B.

**Theorem 4.** *Suppose that $\|X\|_2 \leq 1$ and $\sup_{y \in [-1,1]} |\partial_y \ell(y, Y)| \leq 1$ almost surely, $\|w\|_2 \leq 1$, and that $\ell$ is $\mu$-strongly convex in its first argument. Then, with probability at least $1 - \delta$,*

$$R\left(\frac{1}{T}\sum_{t=1}^{T} w_t\right) - R(w) = O\left(\frac{d \log(T) + \log(1/\delta)}{\mu T}\right),$$

*where $w_t$ are given by Algorithm 2 with $\eta = \frac{\mu}{4}$, $\sigma = 1$, $Z = 1$, $G = 1$, $\kappa_t = 1$, and feedback $g_t = \ell_t'(\langle w_t, X_t \rangle)$ for $t = 1, \ldots, T$.*

# 5 Corrupted Feedback

In this section, we study a setting where the loss derivatives $\ell_t'(\widehat{y}_t)$ observed by the learner at each round $t$ may be adversarially corrupted by unknown additive constants $c_t$, and we are interested in the best possible dependence on $c_1, \ldots, c_T$ in the regret bound. To better explain our setting, we start with the following example.

**Example 3.** *In the online regression setting, suppose that $\ell_t(\widehat{y}_t) = (\widehat{y}_t - y_t)^2$ for all t, but the learner observes corrupted outcomes $y_t - c_t/2$. Hence, the squared loss derivative computed by the learner is $2(\widehat{y}_t - y_t + c_t/2) = \ell_t'(\widehat{y}_t) + c_t$, which can be handled by the algorithms developed in this section.*

Several variants of this setting have been studied in prior work, see for example [Lykouris et al., 2018, Amir et al., 2020, Zimmert and Seldin, 2021, Ito, 2021] and the references therein. The main difference between our setting and these previous settings is that we assume our losses to be strongly convex and the environment is not necessarily stochastic. Although the results in this section are rather straightforward corollaries of our bounds, we believe that it is instructive to provide some explicit results. All proofs of the results in this section are postponed to Appendix C.

Our first result shows the performance of Algorithm 1 in the setup with corrupted gradients. The proof follows from observing that $\ell_t'(\widehat{y}_t)(\widehat{y}_t - y_t^\star) = (\ell_t'(\widehat{y}_t) + c_t)(\widehat{y}_t - y_t^\star) - c_t(\widehat{y}_t - y_t^\star)$ and that for any $\lambda > 0$, the inequality $|c_t(\widehat{y}_t - y_t^\star)| \leq \frac{c_t^2}{\lambda} + \frac{\lambda}{4}(\widehat{y}_t - y_t^\star)^2$ holds. The $\frac{\lambda}{4}(\widehat{y}_t - y_t^\star)^2$ term can be compensated for by the negative $\frac{\mu}{2}(\widehat{y}_t - y_t^\star)^2$ appearing in Lemma 1, leading to a $\sum_{t=1}^{T} c_t^2$ additive term in the regret bound. In particular, our result implies that as long as $\sum_{t=1}^{T} c_t^2$ is of order $O(\log K)$, the same regret bound as if the losses were not corrupted. The formal statement can be found in Theorem 5 below.

**Theorem 5.** *Fix an arbitrary sequence $\ell_1, \ldots, \ell_T$ of $\mu$-strongly convex differentiable losses and corruptions $c_1, \ldots, c_T \in \mathbb{R}$. Then the predictions $\widehat{y}_t$ of Algorithm 1 run with inputs $M \geq$*

$\max_t |\ell'_t(\widehat{y}_t) + c_t|$, $\eta = \frac{\mu}{4}$, *feedback* $g_t = \ell'_t(\widehat{y}_t) + c_t$, *and* $\kappa_t = 1$ *satisfy*

$$\mathcal{R}_T \leq \frac{8M^2 \log(K)}{\mu} + \sum_{t=1}^{T} \frac{c_t^2}{\mu} - \frac{\mu}{8} \sum_{t=1}^{T} (\widehat{y}_t - y_t(i^\star))^2$$

*provided* $\max_i \max_t |\widehat{y}_t - y_t(i)| \leq M$.

Next we prove an analog of Theorem 5 in the online regression setup.

**Theorem 6.** *Fix an arbitrary sequence* $\ell_1, \ldots, \ell_T$ *of* $\mu$-*strongly convex differentiable losses and corruptions* $c_1, \ldots, c_T \in \mathbb{R}$. *Then the predictions* $\widehat{y}_t$ *of Algorithm 2 run with inputs* $\eta = \frac{\mu}{8}$, $\sigma = D^2$, $G \geq \max_t |\ell'_t(\widehat{y}_t) + c_t|$, $Z > 0$, *feedback* $g_t = \ell'_t(\widehat{y}_t) + c_t$, *and* $\kappa_t = 1$, *satisfy*

$$\mathcal{R}_T \leq \frac{4dG^2}{\mu} \log \left( 1 + \frac{TD^2\mu^2 \max_t \|\boldsymbol{x}_t\|_2^2}{2d} \right) + \frac{4G^2}{\mu} + \sum_{t=1}^{T} \frac{c_t^2}{\mu} - \frac{\mu}{8} \sum_{t=1}^{T} (\widehat{y}_t - y_t^\star)^2,$$

*for any* $\boldsymbol{x}_1, \ldots, \boldsymbol{x}_T \in \mathbb{R}^d$, *and for any* $\boldsymbol{u} \in \mathcal{W}_T \equiv \bigcap_{t=1}^{T} \{\boldsymbol{w} : |\langle \boldsymbol{w}, \boldsymbol{x}_t \rangle| \leq Z\}$ *such that* $\|\boldsymbol{u}\|_2 \leq D$ *and* $y_t^\star = \langle \boldsymbol{u}, \boldsymbol{x}_t \rangle$ *for all* $t \geq 1$.

## 6 Selective Sampling

We consider a variant of the selective sampling setting—see, e.g., [Atlas et al., 1989, Freund et al., 1997, Cesa-Bianchi et al., 2003, 2006, Orabona and Cesa-Bianchi, 2011]—where the learner has access to the expert predictions (or, equivalently, to feature vectors), but can observe its own loss only upon request. The goal is to trade off the number of loss requests with regret guarantees. We show that if the variance is high, with only a fraction of all losses requested we obtain the same guarantee (in expectation and up to constants) as when all losses are requested.

Let $o_t = 1$ with probability $q_t$ and $o_t = 0$ with probability $1 - q_t$. In each round, if $o_t = 1$ the loss $\ell_t$ at round $t$ is requested, and we use the loss estimator $\frac{o_t}{q_{t-1}} \ell_t$ to update. Note that this is not the importance weighted estimator, as $\mathbb{E}_{t-1}[\frac{o_t}{q_{t-1}}] = \frac{q_t}{q_{t-1}}$. The reason for choosing this particular loss estimator is that we have better control of the range of the loss which allows us to tune $\kappa_t$ in Algorithm 1 accordingly. The probability of requesting a loss is

$$q_t = \min \left\{ 1, \beta \bigg/ \sqrt{\min_i \sum_{s=1}^{t} (\widehat{y}_s - y_s(i))^2} \right\}, \tag{5}$$

where $\beta > 0$ is chosen by the learner. Our result for the selective sampling setting can be found in Theorem 7 below, whose statement is implied by Theorem 11 in Appendix D. Theorem 7 implies the following: if $\beta = O(\mu^{-3/2} \log(K))$, then with only an expected number $\sum_{t=1}^{T} q_t$ of loss requests, we obtain (up to constants) the same regret guarantee as we would have obtained if we had requested all losses. With this particular choice of $\beta$, $q_t < 1$ as soon the bound in Lemma 1 becomes negative. In other words, when the variance is high we only need a fraction of the losses to recover the worst-case optimal regret bound (in expectation). A similar result can be obtained in the regression setting, see Appendix D.1.

**Theorem 7.** *Fix an arbitrary sequence* $\ell_1, \ldots, \ell_T$ *of* $\mu$-*strongly convex differentiable losses. Provided* $\max_i \max_t |\widehat{y}_t - y_t(i)| \leq 1$, *the predictions* $\widehat{y}_t$ *of Algorithm 1 run with inputs* $M = 1 \geq \max_t |\ell'_t(\widehat{y}_t)|$, $\eta = \frac{\mu}{4}$, *feedback* $g_t = \frac{o_t}{q_{t-1}} \ell'_t(\widehat{y}_t)$, *and* $\kappa_t = q_t$ *satisfy*

$$\mathbb{E} \left[ \sum_{t=1}^{T} (\ell_t(\widehat{y}_t) - \ell_t(y_t^\star)) \right] = O \left( \frac{\log(K)}{\mu} + \frac{\log(K)^2}{\mu^3 \beta^2} \right).$$

## 7 Further Extensions

In Appendix G we present another application of Lemma 1. Namely, we show that we may restart Algorithm 1 for free whenever the regret becomes negative. This gives us a regret bound where we compete with a new expert after each restart, which is a stronger notion of regret than when we compete with a fixed expert in all rounds.

The ideas of Lemma 1 also naturally extend beyond online learning with strongly convex losses. We discuss one such extension to the online prediction with expert advice setting. The online prediction with abstention setting was introduced by Neu and Zhivotovskiy [2020] and proceeds as follows. In each round $t = 1, \ldots, T$ the learner receives expert predictions $y_t(i) \in \{-1, 1\}$, $i = 1, \ldots, K$ and the learner can then either predict $\widetilde{y}_t \in \{-1, 1\}$ or abstain from prediction. If the learner predicts with $\widetilde{y}_t$, the learner suffers the binary loss $\ell_t(\widetilde{y}_t) = \mathbb{1}[\widetilde{y}_t \neq y_t]$, where $y_t \in \{-1, 1\}$. If the learners abstains from prediction, the learner suffers abstention cost $\rho \in [0, \frac{1}{2})$. Let $a_t = 1$ for prediction and $a_t = 0$ for abstention. The total loss of the learner is therefore equal to

$$\sum_{t=1}^{T} (a_t \mathbb{1}[\widetilde{y}_t \neq y_t] + (1 - a_t)\rho).$$

Assume that the prediction strategy is random is a sense that $a_t$ are Bernoulli random variables whose means might depend on previous observations. The work of Neu and Zhivotovskiy [2020] shows that there is a randomized prediction strategy such that for any data generating mechanism it holds that

$$\mathbb{E}\left[\sum_{t=1}^{T} (a_t \mathbb{1}[\widetilde{y}_t \neq y_t] + (1 - a_t)\rho)\right] - \sum_{t=1}^{T} \mathbb{1}[y_t^\star \neq y_t] = O\left(\frac{\log K}{1 - 2\rho}\right), \tag{6}$$

independently of $T$, where the expectation is taken with respect to the randomness of $a_t$; here $y_t^\star = y_t(i^\star)$ and $i^\star = \operatorname{argmin}_i \sum_{t=1}^{T} \mathbb{1}[y_t(i) \neq y_t]$. Although it was shown that the randomization is necessary to achieve the regret bound (6), it is unclear if the same regret bound can be achieved with high probability with respect to the randomization of the algorithm. In Appendix F, we answer this question using the techniques we developed and provide a randomized algorithm such that, with probability at least $1 - \delta$,

$$\sum_{t=1}^{T} (a_t \mathbb{1}[\widetilde{y}_t \neq y_t] + (1 - a_t)\rho) - \sum_{t=1}^{T} \mathbb{1}[y_t^\star \neq y_t] = O\left(\frac{\log K + \log(1/\delta)}{1 - 2\rho}\right).$$

In Appendix F we prove Lemma 7, which is the analog of Lemma 1 for the abstention setting. The equivalent of the negative term in Lemma 1 is used to compensate for the variance of the high-probability statement, which allows us to recover the above bound. This also implies that our other applications of Lemma 1 can be exported to the online learning with abstention setting.

## Acknowledgments and Disclosure of Funding

Dirk van der Hoeven and Nicolò Cesa-Bianchi gratefully acknowledge partial support from the MIUR PRIN grant Algorithms, Games, and Digital Markets (ALGADIMAR), the EU Horizon 2020 ICT-48 research and innovation action under grant agreement 951847, project ELISE (European Learning and Intelligent Systems Excellence), and the project "One Health Action Hub: University Task Force for the resilience of territorial ecosystems" funded by Università degli Studi di Milano. A significant part of this work was done while Nikita Zhivotovskiy was at ETH, Zürich. During this period, Nikita Zhivotovskiy was funded in part by ETH Foundations of Data Science (ETH-FDS).

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
