# A    Details of Section 3 (Our Algorithms)

We restate Lemma 2, after which we prove its result.

**Lemma 2.** *For all $g_1, \ldots, g_T \in [-M, M]$ and $\kappa_0 = \kappa_1 \geq \kappa_2 \cdots \geq \kappa_T$ such that $\kappa_t \in (0, 1]$, the predictions $\widehat{y}_t$ of Algorithm 1 run with input $M$ and $\eta \in (0, \frac{1}{2}]$ satisfy*

$$\sum_{t=1}^{T} (\widehat{y}_t - y_t(i))g_t \leq \frac{M^2 \log(K)}{\kappa_T \eta} + \eta \sum_{t=1}^{T} \kappa_{t-1}(\widehat{y}_t - y_t^\star)^2$$

*provided $\max_i |y_t(i) - y_t^\star| \leq M$ for all $t \geq 1$.*

*Proof.* We start by observing that the vector of weights $\boldsymbol{p}_t$ is obtained by running lazy EWA with learning rate $\kappa_t$ on the (signed) surrogate losses $\widetilde{\ell}_t(i) = \gamma(y_t(i) - \widehat{y}_t)g_t + \kappa_t(\gamma(y_t(i) - \widehat{y}_t)g_t)^2$. We use $\kappa_0 = \kappa_1$. Thus, by [Van der Hoeven et al., 2018, Lemma 1] we have that, for any $i^\star \in [K]$,

$$
\begin{aligned}
&\sum_{t=1}^{T} (\mathbb{E}_{i \sim \boldsymbol{p}_t}[\widetilde{\ell}_t(i)] - \widetilde{\ell}_t(i^\star)) \\
&\leq \frac{\log(K)}{\kappa_T} + \sum_{t=1}^{T} \left( \mathbb{E}_{i \sim \boldsymbol{p}_t}[\widetilde{\ell}_t(i)] + \frac{1}{\kappa_{t-1}} \log \mathbb{E}_{i \sim \boldsymbol{p}_t} \left[ \exp(-\kappa_{t-1}\widetilde{\ell}_t(i)) \right] \right).
\end{aligned}
\tag{7}
$$

Now, using that $\exp(x - x^2) \leq 1 + x$ for $|x| \leq \frac{1}{2}$ (Lemma 2.4 by Cesa-Bianchi and Lugosi [2006]) and the fact that $|\gamma(y_t(i) - \widehat{y}_t)g_t| \leq \frac{1}{2}$ due to our choice of $\gamma$, we find that

$$
\begin{aligned}
&\sum_{t=1}^{T} (\mathbb{E}_{i \sim \boldsymbol{p}_t}[\widetilde{\ell}_t(i)] - \widetilde{\ell}_t(i^\star)) \\
&\leq \log(K) + \sum_{t=1}^{T} \left( \mathbb{E}_{i \sim \boldsymbol{p}_t}[\widetilde{\ell}_t(i)] + \frac{1}{\kappa_{t-1}} \log \mathbb{E}_{i \sim \boldsymbol{p}_t}[1 + \gamma\kappa_{t-1}(\widehat{y}_t - y_t(i))g_t] \right) \\
&= \log(K) + \sum_{t=1}^{T} \mathbb{E}_{i \sim \boldsymbol{p}_t}[\widetilde{\ell}_t(i)] \,,
\end{aligned}
$$

where the equality is due the fact that since $\widehat{y}_t = \mathbb{E}_{i \sim \boldsymbol{p}_t}[y_t(i)]$, we have that $\mathbb{E}_{i \sim \boldsymbol{p}_t}[\gamma(\widehat{y}_t - y_t(i))g_t] = 0$. This also implies that $\mathbb{E}_{i \sim \boldsymbol{p}_t}[\widetilde{\ell}_t(i)] = \mathbb{E}_{i \sim \boldsymbol{p}_t}[\kappa_{t-1}(\gamma(\widehat{y}_t - y_t(i))g_t)^2]$. Thus, we may write

$$
\begin{aligned}
&\mathbb{E}_{i \sim \boldsymbol{p}_t}[\widetilde{\ell}_t(i)] - \widetilde{\ell}_t(i^\star) \\
&= \mathbb{E}_{i \sim \boldsymbol{p}_t}[\kappa_{t-1}(\gamma(\widehat{y}_t - y_t(i))g_t)^2] + \gamma(\widehat{y}_t - y_t(i))g_t - \kappa_{t-1}(\gamma(\widehat{y}_t - y_t(i))g_t)^2.
\end{aligned}
$$

Combining with the above and reordering we find

$$\sum_{t=1}^{T} \gamma(\widehat{y}_t - y_t(i))g_t \leq \frac{\log(K)}{\kappa_T} + \sum_{t=1}^{T} \kappa_{t-1}(\gamma(\widehat{y}_t - y_t(i))g_t)^2 \tag{8}$$

After dividing both sides by $\gamma = \frac{\eta}{M^2}$, this gives us

$$
\begin{aligned}
\sum_{t=1}^{T} (\widehat{y}_t - y_t(i))g_t &\leq \frac{M^2 \log(K)}{\kappa_T \eta} + \frac{\eta}{M^2} \sum_{t=1}^{T} \kappa_{t-1}((\widehat{y}_t - y_t(i))g_t)^2 \\
&\leq \frac{M^2 \log(K)}{\kappa_T \eta} + \eta \sum_{t=1}^{T} \kappa_{t-1}(\widehat{y}_t - y_t(i^\star))^2 \,.
\end{aligned}
$$

completing the proof.                                                                                      $\square$

We now restate Theorem 1 and provide its proof.

**Theorem 1.** *Consider the squared loss and two experts $y_t(1) = 0$ and $y_t(2) = 1$ for all $t \geq 1$. Let $\widehat{y}_t^{EWA}$ be the EWA predictions. There is a sequence $y_1, y_2, \ldots$ such that $y_t \in [0, 1]$, $t \geq 1$ and, for large enough $T$, the regret of EWA with $\eta = \frac{1}{2}$ satisfies $-12 \log T \leq \mathcal{R}_T \leq 2 \log 2$ and, at the same time,*

$$\sum_{t=1}^{T} (\widehat{y}_t^{EWA} - y_t^\star)^2 \geq T/2 \,.$$

*Proof.* The proof uses a construction similar to one used in Audibert [2007]. Our idea is to show that for some environments the output of EWA is close to the *follow the leader* prediction. This can lead to large variance when we follow a wrong expert for most of the rounds. For $T$ large enough consider the following sequence:

$$y_t = \begin{cases} 3/4, & \text{if } t \leq 4\lceil \log T \rceil, \\ 1/2, & \text{if } 4\lceil \log T \rceil < t < T - 8\lceil \log T \rceil, \\ 1/4, & \text{if } T - 8\lceil \log T \rceil \leq t \leq T. \end{cases}$$

Fix $\eta = 1/2$. Since $y_t, y_t(1), y_t(2) \in [0, 1]$, and the squared loss is $1/2$-exp-concave on this domain (see Appendix E) we have $\mathcal{R}_T \leq 2 \log 2$.

Next, we show the lower bound. Since $y_t = 1/4$ appears more frequently in the sequence, we have that $y_t(1) = 0$ is the prediction of the best expert. However, until the last $8\lceil \log T \rceil$ rounds, that is, for any $4\lceil \log T \rceil < t' < T - 8\lceil \log T \rceil$ the EWA algorithm puts most of its weight on the second expert predicting $y_t(2) = 1$. At the same time, both experts suffer the same loss when $y_t = 1/2$. Formally, for any such $t'$ we have

$$\sum_{t=1}^{t'} (y_t - y_t(1))^2 - \sum_{t=1}^{t'} (y_t - y_t(2))^2 = 4\lceil \log T \rceil (9/16 - 1/16) = 2\lceil \log T \rceil.$$

Therefore, for the same $t'$, the weight of the first expert in the EWA prediction with $\eta = 1/2$ is

$$p_{t'}(1) = \frac{\exp\left(-\sum\limits_{t=1}^{t'} (y_t - y_t(1))^2/2\right)}{\sum\limits_{i=1}^{2} \exp\left(-\sum\limits_{t=1}^{t'} (y_t - y_t(i))^2/2\right)} = \frac{1}{1 + \exp(\lceil \log T \rceil)} \leq \frac{1}{1+T} \,.$$

Thus, we have

$$\widehat{y}_{t'}^{EWA} = p_{t'}(1)y_{t'}(1) + p_{t'}(2)y_{t'}(2) \geq \frac{T}{T+1} \,.$$

We are ready to bound the regret. Our idea will be just to use the boundedness of the loss when $y_t \in \{1/4, 3/4\}$ and compute the regret over remaining rounds. Using elementary algebra, we have

$$\mathcal{R}_T \geq (T - 12\lceil \log T \rceil) \cdot \left( \left( \frac{T}{T+1} - 1/2 \right)^2 - (1/2)^2 \right) - 12\lceil \log T \rceil (3/4)^2$$

$$\geq \frac{-T(T - 12\lceil \log T \rceil)}{(T+1)^2} - 12(3/4)^2\lceil \log T \rceil \geq -12 \log T \,,$$

for all $T \geq 4$. At the same time, the following variance bound holds

$$\sum_{t=1}^{T} (\widehat{y}_t^{EWA} - y_t(1))^2 \geq \sum_{t=4\lceil \log T \rceil + 1}^{T-8\lceil \log T \rceil - 1} (\widehat{y}_t^{EWA} - y_t(1))^2 \geq (T - 12\lceil \log T \rceil - 2)\left( \frac{T}{T+1} \right)^2 \geq \frac{T}{2} \,,$$

provided that $T > 150$. The claim follows. $\qquad\square$

Here we restate Lemma 3, after which we prove it.

**Lemma 3.** *For all $g_1, \ldots, g_T \in \mathbb{R}$ and $\kappa_1 \geq \cdots \geq \kappa_T \in (0, 1]$, the predictions $\widehat{y}_t$ of Algorithm 2 run with inputs $\eta > 0$, $\sigma = D^2$, $G \geq \max_t |g_t|$, and $Z > 0$*

$$\sum_{t=1}^{T} (\widehat{y}_t - y_t(\boldsymbol{u})) g_t \leq \frac{dG^2}{2\kappa_T \eta} \log\left(1 + D^2 \eta^2 \left(\max_{t=1,\ldots,T} \|\boldsymbol{x}_t\|_2^2\right) \frac{T}{d}\right) + \frac{G^2}{2\eta} + \eta \sum_{t=1}^{T} \kappa_t (\widehat{y}_t - y_t(\boldsymbol{u}))^2 \,,$$

*for any $\boldsymbol{x}_1, \ldots, \boldsymbol{x}_T \in \mathbb{R}^d$, and for any $\boldsymbol{u} \in \mathcal{W}_T \equiv \bigcap_{t=1}^{T} \{\boldsymbol{w} : |\langle \boldsymbol{w}, \boldsymbol{x}_t \rangle| \leq Z\}$ such that $\|\boldsymbol{u}\|_2 \leq D$.*

*Proof.* We start by observing that $\boldsymbol{w}_t$ is the mean of continuous exponential weights with a Gaussian prior and learning rate 1 on (signed) surrogate losses $\widetilde{\ell}_t^{\mathrm{or}}(\boldsymbol{w}) = \langle \boldsymbol{w} - \boldsymbol{w}_t, \boldsymbol{z}_t \rangle + \kappa_t (\boldsymbol{w} - \boldsymbol{w}_t)^\top \boldsymbol{z}_t \boldsymbol{z}_t^\top (\boldsymbol{w} - \boldsymbol{w}_t)$, see [Van der Hoeven et al., 2018, Section 4]. Thus, for any $\boldsymbol{u} \in \mathcal{W}_T$, by Van der Hoeven et al. [2018, Theorem 5] we have that

$$\sum_{t=1}^{T} \left(\widetilde{\ell}_t^{\mathrm{or}}(\boldsymbol{w}_t) - \widetilde{\ell}_t^{\mathrm{or}}(\boldsymbol{u})\right) \leq \frac{\|\boldsymbol{u}\|_2^2}{2\sigma} + \frac{1}{2} \sum_{t=1}^{T} \boldsymbol{z}_t^\top \Sigma_{t+1} \boldsymbol{z}_t \,.$$

Using that $\kappa_1 \geq \kappa_2 \cdots \geq \kappa_T \in (0, 1]$ and the Sherman-Morrison formula to compute the inverse we find that

$$2\boldsymbol{z}_t^\top \left(2 \sum_{s=1}^{t} \kappa_s \boldsymbol{z}_s \boldsymbol{z}_s^\top + \frac{1}{\sigma} I\right)^{-1} \boldsymbol{z}_t \leq \frac{1}{\kappa_T} 2\boldsymbol{z}_t^\top \left(2 \sum_{s=1}^{t} \boldsymbol{z}_s \boldsymbol{z}_s^\top + \frac{1}{\sigma} I\right)^{-1} \boldsymbol{z}_t$$

$$= \frac{1}{\kappa_T} \left(2\boldsymbol{z}_t^\top \Sigma_t \boldsymbol{z}_t - \frac{(2\boldsymbol{z}_t^\top \Sigma_t \boldsymbol{z}_t)^2}{1 + 2\boldsymbol{z}_t^\top \Sigma_t \boldsymbol{z}_t}\right)$$

$$= \frac{1}{\kappa_T} \left(1 - \frac{1}{1 + 2\boldsymbol{z}_t^\top \Sigma_t \boldsymbol{z}_t}\right)$$

$$\leq \frac{1}{\kappa_T} \log\left(1 + 2\boldsymbol{z}_t^\top \Sigma_t \boldsymbol{z}_t\right) = \frac{1}{\kappa_T} \log\left(\frac{\mathrm{Det}(\Sigma_{t+1})}{\mathrm{Det}(\Sigma_t)}\right) \,,$$

where the second inequality is due to the fact that $1 - 1/x \leq \log(x)$ for $x > 0$ and the final equality can be found on, for example, in Meyer [2001, page 475]. Thus, we have that

$$\sum_{t=1}^{T} \boldsymbol{z}_t^\top \Sigma_{t+1} \boldsymbol{z}_t \leq \frac{1}{\kappa_T} \sum_{t=1}^{T} \frac{1}{2} \log\left(\frac{\mathrm{Det}(\Sigma_{t+1})}{\mathrm{Det}(\Sigma_t)}\right)$$

$$\leq \frac{1}{2\kappa_T} \log\left(\frac{\mathrm{Det}(\Sigma_{T+1})}{\mathrm{Det}(\Sigma_1)}\right) = \frac{1}{2\kappa_T} \log \mathrm{Det}\left(I + \sigma \sum_{t=1}^{T} 2\boldsymbol{z}_t \boldsymbol{z}_t^2\right) \,.$$

Now, following the proof and discussion of Cesa-Bianchi and Lugosi [2006, Theorem 11.2] we have that

$$\log \mathrm{Det}\left(I + \sigma^2 \sum_{t=1}^{T} 2\boldsymbol{z}_t \boldsymbol{z}_t^2\right) \leq d \log\left(1 + 2\sigma \left(\max_t \|\boldsymbol{z}_t\|_2^2\right) \frac{T}{d}\right)$$

$$\leq d \log\left(1 + 2\sigma \gamma^2 \left(\max_t \|\boldsymbol{x}_t g_t\|_2^2\right) \frac{T}{d}\right) \,.$$

By combining the above we find

$$\sum_{t=1}^{T} \langle \boldsymbol{w}_t - \boldsymbol{u}, \boldsymbol{x}_t g_t \rangle = \frac{1}{\gamma} \sum_{t=1}^{T} (\langle \boldsymbol{w}_t - \boldsymbol{u}, \boldsymbol{z}_t \rangle - \kappa_t \langle \boldsymbol{w}_t - \boldsymbol{u}, \boldsymbol{z}_t \rangle^2) + \sum_{t=1}^{T} \gamma \kappa_t (\langle \boldsymbol{w}_t - \boldsymbol{u}, \boldsymbol{x}_t g_t \rangle)^2$$

$$\leq \frac{\|\boldsymbol{u}\|_2^2}{2\sigma\gamma} + \frac{d}{2\gamma\kappa_T} \log\left(1 + 2\sigma\gamma^2 \left(\max_t \|\boldsymbol{x}_t g_t\|_2^2\right) \frac{T}{d}\right) + \sum_{t=1}^{T} \gamma \kappa_t (\langle \boldsymbol{w}_t - \boldsymbol{u}, \boldsymbol{x}_t g_t \rangle)^2 \,.$$

$$(9)$$

Now, we continue by using that $\langle \boldsymbol{w}_t - \boldsymbol{u}, \boldsymbol{x}_t \rangle = \widehat{y}_t - y_t(\boldsymbol{u}), \gamma = \frac{\eta}{G^2}, \sigma = D^2$, and that $\|\boldsymbol{u}\|_2 \leq D$

$$\sum_{t=1}^{T} (\widehat{y}_t - y_t(\boldsymbol{u})) g_t$$

$$\leq \frac{1}{2\gamma} + \frac{dG^2}{2\kappa_T \eta} \log\left(1 + 2D^2\gamma^2 \Big(\max_t \|\boldsymbol{x}_t g_t\|_2^2\Big)\frac{T}{d}\right) + \sum_{t=1}^{T} \gamma\kappa_t((\widehat{y}_t - y_t(\boldsymbol{u}))g_t)^2$$

$$\leq \frac{G^2}{2\eta} + \frac{dG^2}{2\kappa_T \eta} \log\left(1 + 2D^2\eta^2 \Big(\max_t \|\boldsymbol{x}_t\|_2^2\Big)\frac{T}{d}\right) + \eta \sum_{t=1}^{T} \kappa_t(\widehat{y}_t - y_t(\boldsymbol{u}))^2,$$

where we used the assumption that $g_t^2 \leq G^2$, which completes the proof. $\qquad\square$

# B   Details of Section 4 (Statistical Learning)

To directly use our results obtained in the general online setting, we set the following notation for the rest of the section:

$$\ell_t(\cdot) = \ell(\cdot, Y_t), \quad y_t(f) = f(X_t), \quad y_t^\star = f^\star(X_t) \quad \text{and} \quad \widehat{y}_t = \widehat{f}_t(X_t), \tag{10}$$

where $\widehat{f}_t$ is a statistical estimator constructed based on $(X_1, Y_1), \ldots, (X_{t-1}, Y_{t-1})$.

Let $r_t = \ell_t(\widehat{y}_t) - \ell_t(y_t^\star)$. To prove our high-probability bounds, we are interested in controlling $\sum_{t=1}^{T} \mathbb{E}_{t-1}[r_t]$. As we mentioned, the challenge in obtaining high-probability bounds comes from bounding $\mathbb{E}_{t-1}[r_t] - r_t$, which may be of order $\sqrt{T}$ in the worst case due to the variance of $r_t$. The negative term is our regret bounds is used to control the variance term.

The following two versions of Freedman's inequality for martingales appear explicitly in [Beygelzimer et al., 2011, Theorem 1] and [Rakhlin et al., 2012, Lemma 3]. We use them to prove Lemmas 5 and 6, which in turn are used to prove Theorems 8, 9, and 10.

**Lemma 4** (Versions of Freedman's inequality). *Let $X_1, \ldots, X_T$ be a martingale difference sequence adapted to a filtration $(\mathcal{F}_i)_{i \leq T}$. That is, in particular, $\mathbb{E}_{-1}[X_t] = 0$. Suppose that $|X_t| \leq R$ almost surely. Then for any $\delta \in (0, 1), \lambda \in [0, 1/R]$, with probability at least $1 - \delta$, it holds that*

$$\sum_{t=1}^{T} X_t \leq \lambda(e-2) \sum_{t=1}^{T} \mathbb{E}_{t-1}[X_t^2] + \frac{\log(1/\delta)}{\lambda}. \tag{11}$$

*Moreover, if $\delta \in (0, 1/2), T \geq 4$, then uniformly over all $s \leq T$, with probability at least $1 - \delta$, it holds that*

$$\sum_{t=1}^{s} X_t \leq 4\sqrt{\sum_{t=1}^{s} \mathbb{E}_{t-1}[X_t^2] \log(\log(T)/\delta)} + 2R\log(\log(T)/\delta). \tag{12}$$

**Lemma 5.** *Under the notation* (10) *suppose that $\max_t \max\{|\widehat{y}_t|, |y_t^\star|\} \leq \frac{1}{2}M$ almost surely and that $\max_t |\ell_t'(y)| \leq M$ almost surely for all $y$ such that $|y| \leq \frac{1}{2}M$. Suppose that $\widehat{y}_1, \ldots, \widehat{y}_T$ satisfy* (2) *with $\eta = \frac{\mu}{4}$. Then, with probability at least $1 - \delta$, it holds that*

$$\sum_{t=1}^{T} \mathbb{E}_{t-1}[\ell_t(\widehat{y}_t) - \ell_t(y_t^\star)] \leq \frac{8C_T}{\mu} + B_T + \log(1/\delta) \min\left\{ \frac{1}{(2 + \frac{\mu}{2})M^2}, \frac{\mu}{6M^2}\left(1 + \frac{\mu^2}{16}\right)^{-1} \right\}^{-1}.$$

*Proof.* Let $v_t = (\widehat{y}_t - y_t^\star)^2$ and let $r_t = \ell_t(\widehat{y}_t) - \ell_t(y_t^\star)$. By convexity and the assumptions on $y_t^\star, \widehat{y}_t$, and $\ell_t'$ we have that $|\ell_t(\widehat{y}_t) - \ell_t(y_t^\star)| \leq |\widehat{y}_t - y_t^\star|M \leq M^2$. This implies that

$$\left| r_t - \frac{\mu}{4}v_t \right| \leq |r_t| + \frac{\mu}{4}v_t \leq M^2 + \frac{\mu}{4}M^2.$$

Thus, by equation (11) we have that, for $\lambda \in \left[0, \frac{1}{2\left(M^2 + \frac{\mu}{4}M^2\right)}\right]$, with probability at least $1 - \delta$,

$$\sum_{t=1}^{T} \mathbb{E}_{t-1}\left[r_t + \frac{\mu}{4}v_t\right] - \frac{\mu}{4}v_t - r_t$$

$$\leq \lambda(e-2) \sum_{t=1}^{T} \mathbb{E}_{t-1}\left[\left(\mathbb{E}_{t-1}\left[r_t + \frac{\mu}{4}v_t\right] - \frac{\mu}{4}v_t - r_t\right)^2\right] + \frac{\log(1/\delta)}{\lambda}$$

$$\leq \lambda(e-2) \sum_{t=1}^{T} \mathbb{E}_{t-1}\left[\left(\frac{\mu}{4}v_t + r_t\right)^2\right] + \frac{\log(1/\delta)}{\lambda} \ .$$

where we used that $\mathbb{E}[(X - \mathbb{E}[X])^2] \leq \mathbb{E}[X^2]$. Since $r_t^2 \leq (y_t^\star - \widehat{y}_t)^2 M^2$ we have that

$$\mathbb{E}_{t-1}\left[\left(r_t + \frac{\mu}{4}v_t\right)^2\right] \leq 2\,\mathbb{E}_{t-1}[r_t^2] + \frac{\mu^2}{8}\,\mathbb{E}[v_t^2]$$

$$\leq \mathbb{E}_{t-1}\left[2(y_t^\star - \widehat{y}_t)^2 M^2 + \frac{\mu^2 M^2}{8}(y_t^\star - \widehat{y}_t)^2\right]$$

$$= 2\left(M^2 + \frac{\mu^2 M^2}{16}\right)\mathbb{E}_{t-1}[v_t] \ .$$

which means that, with probability at least $1 - \delta$,

$$\sum_{t=1}^{T} \mathbb{E}_{t-1}\left[r_t + \frac{\mu}{4}v_t\right] - \frac{\mu}{4}v_t - r_t$$

$$\leq \lambda(e-2) \sum_{t=1}^{T} 2\left(M^2 + \frac{\mu^2 M^2}{16}\right)\mathbb{E}_{t-1}[v_t] + \frac{\log(1/\delta)}{\lambda} \tag{13}$$

$$\leq \lambda \sum_{t=1}^{T} \frac{3}{2}\left(M^2 + \frac{\mu^2 M^2}{16}\right)\mathbb{E}_{t-1}[v_t] + \frac{\log(1/\delta)}{\lambda}$$

Using the guarantee on $\mathcal{R}_T$ in equation (2), Lemma 1, and replacing $\eta$ with $\frac{\mu}{4}$ we have that

$$\sum_{t=1}^{T} \mathbb{E}_{t-1}[r_t]$$

$$= \mathcal{R}_T + \sum_{t=1}^{T}(\mathbb{E}_{t-1}[r_t] - r_t)$$

$$\leq \frac{C_T}{\eta} + \left(\eta - \frac{\mu}{2}\right)\sum_{t=1}^{T} v_t + B_T + \sum_{t=1}^{T}(\mathbb{E}_{t-1}[r_t] - r_t)$$

$$= \frac{C_T}{\eta} + \left(\eta - \frac{\mu}{4}\right)\sum_{t=1}^{T} v_t + B_T - \frac{\mu}{4}\sum_{t=1}^{T}\mathbb{E}_{t-1}[v_t] + \sum_{t=1}^{T}\left(\mathbb{E}_{t-1}[r_t] - r_t + \frac{\mu}{4}(\mathbb{E}_{t-1}[v_t] - v_t)\right)$$

$$= \frac{8C_T}{\mu} + B_T - \frac{\mu}{4}\sum_{t=1}^{T}\mathbb{E}_{t-1}[v_t] + \sum_{t=1}^{T}\left(\mathbb{E}_{t-1}[r_t] - r_t + \frac{\mu}{4}(\mathbb{E}_{t-1}[v_t] - v_t)\right) \ .$$

Thus, by equation (13) we have that, with probability at least $1 - \delta$,

$$\sum_{t=1}^{T} \mathbb{E}_{t-1}[r_t] \leq \frac{8C_T}{\mu} + B_T - \frac{\mu}{4}\sum_{t=1}^{T}\mathbb{E}_{t-1}[v_t] + \lambda\sum_{t=1}^{T}\frac{3}{2}\left(M^2 + \frac{\mu^2 M^2}{16}\right)\mathbb{E}_{t-1}[v_t] + \frac{\log(1/\delta)}{\lambda} \ .$$

Thus, setting $\lambda = \min\left\{\frac{1}{2\left(M^2 + \frac{\mu}{4}M^2\right)}, \frac{\mu}{6}\left(M^2 + \frac{\mu^2 M^2}{16}\right)^{-1}\right\}$ completes the proof. $\square$

**Lemma 6.** *Under the notation* (10) *suppose that* $\max_t \max\{|\widehat{y}_t|, |y_t^\star|\} \le \frac{1}{2}M$ *almost surely and that* $\max_t |\ell_t'(y)| \le M$ *almost surely for all $y$ such that $|y| \le \frac{1}{2}M$. Suppose that $\widehat{y}_1, \ldots, \widehat{y}_S$ satisfy* (2) *with* $\eta = \frac{\mu}{8}$. *Then, for* $\delta \in (0, \frac{1}{2})$, $T \ge 4$, *and uniformly over all $S \le T$, with probability at least* $1 - \delta$, *it holds that*

$$\sum_{t=1}^S \mathbb{E}_{t-1}[\ell_t(\widehat{y}_t) - \ell_t(y_t^\star)] \le \frac{8C_T}{\mu} + B_T - \frac{\mu}{8}\sum_{t=1}^S (\widehat{y}_t - y_t^\star)^2 + M^2 \left(\frac{32}{\mu} + 3\mu + 4\right) \log\left(\frac{\log T}{\delta}\right).$$

*Proof.* Let $v_t = (\widehat{y}_t - y_t^\star)^2$ and $r_t = \ell_t(\widehat{y}_t) - \ell_t(y_t^\star)$. By convexity and the assumptions on $y_t^\star, \widehat{y}_t$, and $\ell_t'$ we have that $|\ell_t(\widehat{y}_t) - \ell_t(y_t^\star)| \le |\widehat{y}_t - y_t^\star|M \le M^2$. This implies that

$$\left|r_t - \frac{\mu}{4}v_t\right| \le |r_t| + \frac{\mu}{4}v_t \le M^2 + \frac{\mu}{4}M^2.$$

Thus, by equation (12) we have that, with probability at least $1 - \delta$

$$\sum_{t=1}^s \mathbb{E}_{t-1}\left[r_t + \frac{\mu}{4}v_t\right] - \frac{\mu}{4}v_t - r_t$$

$$\le 4\sqrt{\sum_{t=1}^s \mathbb{E}_{t-1}\left[\left(\frac{\mu}{4}v_t + r_t\right)^2\right]\log(\log(T)/\delta)} + M^2(4 + \mu)\log(\log(T)/\delta).$$

where we used that $\mathbb{E}[(X - \mathbb{E}[X])^2] \le \mathbb{E}[X^2]$. Since $r_t^2 \le (y_t^\star - \widehat{y}_t)^2 M^2$ we have that

$$\mathbb{E}_{t-1}\left[\left(r_t + \frac{\mu}{4}v_t\right)^2\right] \le 2\mathbb{E}_{t-1}[r_t^2] + \frac{2\mu^2}{16}\mathbb{E}[v_t^2]$$

$$\le \mathbb{E}_{t-1}\left[2(y_t^\star - \widehat{y}_t)^2 M^2 + \frac{2\mu^2 M^2}{16}(y_t^\star - \widehat{y}_t)^2\right] = 2\left(M^2 + \frac{\mu^2 M^2}{16}\right)\mathbb{E}_{t-1}[v_t].$$

which means that with probability at least $1 - \delta$

$$\sum_{t=1}^s \mathbb{E}_{t-1}\left[r_t + \frac{\mu}{4}v_t\right] - \frac{\mu}{4}v_t - r_t$$

$$\le 4\sqrt{\sum_{t=1}^s 2\left(M^2 + \frac{\mu^2 M^2}{16}\right)\mathbb{E}_{t-1}[v_t]\log(\log(T)/\delta) + 2R\log(\log(T)/\delta)} \qquad (14)$$

$$\le \lambda\frac{1}{2}\sum_{t=1}^s \mathbb{E}_{t-1}[v_t] + \frac{16}{\lambda}\left(M^2 + \frac{\mu^2 M^2}{16}\right)\log(\log(T)/\delta) + M^2(4 + \mu)\log(\log(T)/\delta)$$

for any $\lambda > 0$, where in the final inequality we used $\sqrt{ab} = \frac{1}{2}\inf_{\eta > 0}\eta a + \frac{b}{\eta}$ for $a, b \ge 0$. Using the guarantee on $\mathcal{R}_T$ in equation (2), Lemma 1, and replacing $\eta$ with $\frac{\mu}{8}$ we have that

$$\sum_{t=1}^S \mathbb{E}_{t-1}[r_t] = \mathcal{R}_T + \sum_{t=1}^S (\mathbb{E}_{t-1}[r_t] - r_t)$$

$$\le \frac{C_T}{\eta} + \left(\eta - \frac{\mu}{2}\right)\sum_{t=1}^S v_t + B_T + \sum_{t=1}^S (\mathbb{E}_{t-1}[r_t] - r_t)$$

$$= \frac{C_T}{\eta} + \left(\eta - \frac{\mu}{4}\right)\sum_{t=1}^S v_t + B_T - \frac{\mu}{4}\sum_{t=1}^S \mathbb{E}_{t-1}[v_t]$$

$$+ \sum_{t=1}^S \left(\mathbb{E}_{t-1}[r_t] - r_t + \frac{\mu}{4}(\mathbb{E}_{t-1}[v_t] - v_t)\right)$$

$$= \frac{8C_T}{\mu} + B_T - \frac{\mu}{8}\sum_{t=1}^S v_t - \frac{\mu}{4}\sum_{t=1}^S \mathbb{E}_{t-1}[v_t]$$

$$+ \sum_{t=1}^S \left(\mathbb{E}_{t-1}[r_t] - r_t + \frac{\mu}{4}(\mathbb{E}_{t-1}[v_t] - v_t)\right).$$

Thus, by (14) we have that, with probability at least $1 - \delta$,

$$\sum_{t=1}^{S} \mathbb{E}_{t-1}[r_t] \leq \frac{8C_T}{\mu} + B_T - \frac{\mu}{8}\sum_{t=1}^{S} v_t - \frac{\mu}{4}\sum_{t=1}^{S} \mathbb{E}_{t-1}[v_t] + \lambda \frac{1}{2}\sum_{t=1}^{s} \mathbb{E}_{t-1}[v_t]$$
$$+ \frac{16}{\lambda}\left(M^2 + \frac{\mu^2 M^2}{16}\right)\log(\log(T)/\delta) + M^2(4+\mu)\log(\log(T)/\delta)$$

Thus, setting $\lambda = \frac{\mu}{2}$ gives us

$$\sum_{t=1}^{S} \mathbb{E}_{t-1}[r_t] \leq \frac{8C_T}{\mu} + B_T - \frac{\mu}{8}\sum_{t=1}^{S} v_t + M^2\left(\frac{32}{\mu} + 3\mu + 4\right)\log(\log(T)/\delta),$$

which completes the proof. $\qquad\square$

The following Theorem is the detailed statement of Theorem 2 in the main body of the paper.

**Theorem 8.** *Suppose that for all $f \in \mathcal{F}$ $|f(X)| \leq \frac{1}{2}M$ almost surely, that $|\partial_y \ell(y, Y)| \leq M$ almost surely for all $y$ such that $|y| \leq \frac{1}{2}M$, and that $\ell$ is $\mu$-strongly convex in its first argument. Then, with probability at least $1 - \delta$, Algorithm 3 with input parameters $T$, $\mathcal{S} = \frac{8M^2 \log(K)}{\mu} + M^2\left(\frac{32}{\mu} + 3\mu + 4\right)\log(\log(T)/\delta)$, $\eta = \frac{\mu}{8}$, and $M$ guarantees*

$$R(\widehat{f}) - \min_{f \in \mathcal{F}} R(f) \leq \begin{cases} 0 & \text{if } S < T, \\ \frac{M^2}{\mu T}\left(8\log(K) + (32 + 3\mu^2 + 4\mu)\log(\log(T)/\delta)\right) & \text{if } S = T. \end{cases}$$

*Proof.* Convexity of $R$ gives us

$$R(\widehat{f}) \leq \frac{1}{S}\sum_{t=1}^{S} R\left(\sum_{i=1}^{K} p_t(i)f_i\right) = \frac{1}{S}\sum_{t=1}^{S} \mathbb{E}_{t-1}\left[\ell_t\left(\sum_{i=1}^{K} p_t(i)f_i(X_t)\right)\right] = \frac{1}{S}\sum_{t=1}^{S} \mathbb{E}_{t-1}[\ell_t(\widehat{y}_t)].$$

Now, for any fixed $f \in \mathcal{F}$, by Lemma 6 and Lemma 2 we have that, with probability $1 - \delta$, simultaneously for all $S \leq T$,

$$
\begin{aligned}
R(\widehat{f}) - R(f) &\leq \frac{1}{S}\sum_{t=1}^{S} \mathbb{E}_{t-1}[\ell_t(\widehat{y}_t) - \ell_t(y_t(i))] \\
&\leq \frac{\frac{8M^2\log(K)}{\mu} - \frac{\mu}{8}\sum_{t=1}^{S}(\widehat{y}_t - f(X_t))^2 + M^2\left(\frac{32}{\mu} + 3\mu + 4\right)\log(\log(T)/\delta)}{S}.
\end{aligned}
\tag{15}
$$

We split the remainder of the proof into two cases. Either $S < T$ or $S = T$. If $S < T$ then $\mathcal{S} = \frac{8M^2\log(K)}{\mu} + M^2\left(\frac{32}{\mu} + 3\mu + 4\right)\log(\log(T)/\delta) \leq \frac{\mu}{8}\min_i \sum_{t=1}^{S}(\widehat{y}_t - y_t(i))$ and thus

$$R(\widehat{f}) - R(f) \leq 0.$$

To complete the proof, observe that if $S = T$, then from (15) we have that

$$R(\widehat{f}) - R(f) \leq \frac{\frac{8M^2\log(K)}{\mu} + M^2\left(\frac{32}{\mu} + 3\mu + 4\right)\log(\log(T)/\delta)}{T}.$$

The claim follows. $\qquad\square$

The following result is a detailed version of Theorem 3 in the main text.

**Theorem 9.** *Suppose that for all $f \in \mathcal{F}$ $|f(X)| \leq \frac{1}{2}M$ almost surely, that $|\partial_y \ell(y, Y)| \leq M$ almost surely for all $y$ s.t. $|y| \leq \frac{1}{2}M$, and that $\ell$ is $\mu$-strongly convex in its first argument. Then, with probability at least $1 - \delta$, Algorithm 3 with input parameters $T$, $\eta = \frac{\mu}{4}$, $\mathcal{S} = \infty$, and $M$,*

$$R\left(\widehat{f}\right) - \min_{f \in \mathcal{F}} R(f)$$
$$\leq \frac{1}{T}\left(\frac{4M^2\log(K)}{\mu} + \log(1/\delta)\min\left\{\frac{1}{(2+\frac{\mu}{2})M^2}, \frac{\mu}{6M^2}\left(1 + \frac{\mu^2}{16}\right)^{-1}\right\}^{-1}\right).$$

*Proof.* Observe that $\eta < \frac{1}{2}$ by assumption on $\mu$, making it a valid choice. Furthermore, by the choice of $\mathcal{S}$ we have that on termination of Algorithm 3 $S = T$ and thus convexity of $R$ gives us

$$R(\widehat{f}) \leq \frac{1}{T} \sum_{t=1}^{T} R\left(\sum_{i=1}^{K} p_t(i) f_i\right) = \frac{1}{T} \sum_{t=1}^{T} \mathbb{E}_{t-1}\left[\ell_t\left(\sum_{i=1}^{K} p_t(i) f_i(X_t)\right)\right] = \frac{1}{T} \sum_{t=1}^{T} \mathbb{E}_{t-1}[\ell_t(\widehat{y}_t)].$$

Now, for any $f \in \mathcal{F}$, by Lemma 5 and Lemma 2 we have that with probability $1 - \delta$

$$R\left(\widehat{f}\right) - R(f) \leq \frac{1}{T} \sum_{t=1}^{T} \mathbb{E}_{t-1}[\ell_t(\widehat{y}_t) - \ell_t(y_t(i))]$$

$$\leq \frac{1}{T} \left(\frac{4M^2 \log(K)}{\mu} + \log(1/\delta) \min\left\{\frac{1}{(2 + \frac{\mu}{2})M^2}, \frac{\mu}{6M^2}\left(1 + \frac{\mu^2}{16}\right)^{-1}\right\}^{-1}\right),$$

which completes the proof. $\qquad\square$

The following Theorem is a detailed version of Theorem 4.

**Theorem 10.** *Fix $Z > 0$. Suppose that $\ell$ is $\mu-$strongly convex in its first argument and that $\|X_t\|_2 \leq B$ and $\sup_{y \in [-Z, Z]} |\partial \ell(y, T)| \leq G$ almost surely. For any $\boldsymbol{w} \in \{\boldsymbol{w} : |\langle \boldsymbol{w}, x \rangle| \leq Z \text{ for all } x \in \mathcal{X}\}$ such that $\|\boldsymbol{w}\|_2 \leq D$, with probability at least $1 - \delta$*

$$R\left(\frac{1}{T} \sum_{t=1}^{T} \boldsymbol{w}_t\right) - R(\boldsymbol{w}) \leq \frac{1}{T}\left(\frac{8G^2 + \frac{dG^2}{2} \log\left(1 + D^2 \mu^2 B^2 \frac{T}{2d}\right)}{\mu}\right.$$

$$\left. + \log(1/\delta) \min\left\{\frac{1}{(2 + \frac{\mu}{2}) \max\{G, 2Z\}^2}, \frac{\mu}{6 \max\{G, 2Z\}^2}\left(1 + \frac{\mu^2}{16}\right)^{-1}\right\}^{-1}\right),$$

*where $\boldsymbol{w}_t$ are given by Algorithm 2 with $\eta = \frac{\mu}{4}$, $\sigma = D^2$, $G$, $\kappa_t = 1$, and feedback $g_t = \ell'_t(\langle \boldsymbol{w}_t, X_t\rangle)$ for $t = 1, \dots, T$.*

*Proof.* Convexity of $R$ together with $\mathbb{E}_{t-1}[\ell_t(\widehat{y}_t)] = R(\boldsymbol{w}_t)$ gives us

$$R(\bar{\boldsymbol{w}}) \leq \frac{1}{T} \sum_{t=1}^{T} R(\boldsymbol{w}_t) = \frac{1}{T} \sum_{t=1}^{T} \mathbb{E}_{t-1}[\ell_t(\widehat{y}_t)].$$

Let $y_t(\boldsymbol{w}) = \langle \boldsymbol{w}, X_t \rangle$. Let $M = \max\{G, 2Z\}$. Since $\{\boldsymbol{w} : |\langle \boldsymbol{w}, x \rangle| \leq Z \text{ for all } x \in \mathcal{X}\} \subseteq \mathcal{W}_T$, by Lemma 5 and Lemma 3, we have that for any fixed $\boldsymbol{w} \in \{\boldsymbol{w} : \|\boldsymbol{w}\|_2 \leq D \text{ and } |\langle \boldsymbol{w}, X \rangle| \leq Z \text{ for all } x \in \mathcal{X}\}$, with probability at least $1 - \delta$,

$$R(\bar{\boldsymbol{w}}) - R(\boldsymbol{w})$$

$$\leq \frac{1}{T} \sum_{t=1}^{T} (\mathbb{E}_{t-1}[\ell_t(\widehat{y}_t)] - \mathbb{E}_{t-1}[\ell_t(y_t(\boldsymbol{w}))])$$

$$\leq \frac{1}{T}\left(\frac{8C_T}{\mu} + \log(1/\delta) \min\left\{\frac{1}{(2 + \frac{\mu}{2})M^2}, \frac{\mu}{6M^2}\left(1 + \frac{\mu^2}{16}\right)^{-1}\right\}^{-1}\right)$$

$$= \frac{1}{T}\left(\frac{8G^2 + \frac{dG^2}{2} \log\left(1 + D^2 \mu^2 B^2 \frac{T}{2d}\right)}{\mu}\right.$$

$$\left. + \log(1/\delta) \min\left\{\frac{1}{(2 + \frac{\mu}{2})M^2}, \frac{\mu}{6M^2}\left(1 + \frac{\mu^2}{16}\right)^{-1}\right\}^{-1}\right).$$

$\qquad\square$

# C   Details of Section 5 (Corrupted Feedback)

We first restate Theorem 5, after which we prove it.

**Theorem 5.** *Fix an arbitrary sequence $\ell_1, \ldots, \ell_T$ of $\mu$-strongly convex differentiable losses and corruptions $c_1, \ldots, c_T \in \mathbb{R}$. Then the predictions $\widehat{y}_t$ of Algorithm 1 run with inputs $M \geq \max_t |\ell'_t(\widehat{y}_t) + c_t|$, $\eta = \frac{\mu}{4}$, feedback $g_t = \ell'_t(\widehat{y}_t) + c_t$, and $\kappa_t = 1$ satisfy*

$$\mathcal{R}_T \leq \frac{8M^2 \log(K)}{\mu} + \sum_{t=1}^{T} \frac{c_t^2}{\mu} - \frac{\mu}{8} \sum_{t=1}^{T} (\widehat{y}_t - y_t(i^\star))^2$$

*provided $\max_i \max_t |\widehat{y}_t - y_t(i)| \leq M$.*

*Proof.* First, observe that $\eta < \frac{1}{2}$ by assumption on $\mu$, making it a valid choice. Using Lemma 2 and

$$|c_t(\widehat{y}_t - y_t^\star)| \leq \frac{c_t^2}{\lambda} + \frac{\lambda}{4}(\widehat{y}_t - y_t^\star)^2 \tag{16}$$

we obtain

$$\sum_{t=1}^{T} (\widehat{y}_t - y_t^\star)\ell'_t(\widehat{y}_t) = \sum_{t=1}^{T} (\widehat{y}_t - y_t^\star)g_t - \sum_{t=1}^{T} (\widehat{y}_t - y_t^\star)c_t$$

$$\leq \frac{M^2 \log(K)}{\eta} + \sum_{t=1}^{T} \left( \eta(\widehat{y}_t - y_t(i^\star))^2 + \frac{\mu}{4}(\widehat{y}_t - y_t^\star)^2 + \frac{c_t^2}{\mu} \right).$$

We conclude by using Lemma 1:

$$\mathcal{R}_T \leq \frac{M^2 \log(K)}{\eta} + \left( \eta + \frac{\mu}{4} - \frac{\mu}{2} \right) \sum_{t=1}^{T} (\widehat{y}_t - y_t(i^\star))^2 + \frac{1}{\mu} \sum_{t=1}^{T} c_t^2 \,,$$

after which replacing $\eta = \frac{\mu}{8}$ completes the proof. $\qquad\square$

We now restate Theorem 6, after which we prove its result.

**Theorem 6.** *Fix an arbitrary sequence $\ell_1, \ldots, \ell_T$ of $\mu$-strongly convex differentiable losses and corruptions $c_1, \ldots, c_T \in \mathbb{R}$. Then the predictions $\widehat{y}_t$ of Algorithm 2 run with inputs $\eta = \frac{\mu}{8}$, $\sigma = D^2$, $G \geq \max_t |\ell'_t(\widehat{y}_t) + c_t|$, $Z > 0$, feedback $g_t = \ell'_t(\widehat{y}_t) + c_t$, and $\kappa_t = 1$, satisfy*

$$\mathcal{R}_T \leq \frac{4dG^2}{\mu} \log \left( 1 + \frac{TD^2 \mu^2 \max_t \|x_t\|_2^2}{2d} \right) + \frac{4G^2}{\mu} + \sum_{t=1}^{T} \frac{c_t^2}{\mu} - \frac{\mu}{8} \sum_{t=1}^{T} (\widehat{y}_t - y_t^\star)^2,$$

*for any $x_1, \ldots, x_T \in \mathbb{R}^d$, and for any $u \in \mathcal{W}_T \equiv \bigcap_{t=1}^{T} \{w : |\langle w, x_t \rangle| \leq Z\}$ such that $\|u\|_2 \leq D$ and $y_t^\star = \langle u, x_t \rangle$ for all $t \geq 1$.*

*Proof.* As in the proof of Theorem 5, using Lemma 3 and equation (16) we obtain

$$\sum_{t=1}^{T} (\widehat{y}_t - y_t^\star)\ell'_t(\widehat{y}_t) = \sum_{t=1}^{T} (\widehat{y}_t - y_t^\star)g_t - \sum_{t=1}^{T} (\widehat{y}_t - y_t^\star)c_t$$

$$\leq \frac{G^2}{2\eta} + \frac{dG^2}{2\eta} \log \left( 1 + \tfrac{1}{2}D^2 \mu^2 \big( \max_t \|x_t\|_2^2 \big) \frac{T}{d} \right) + \sum_{t=1}^{T} \left( \eta(\widehat{y}_t - y_t^\star)^2 + \frac{c_t^2}{\mu} \right).$$

We continue by using Lemma 1:

$$\mathcal{R}_T \leq \frac{G^2}{2\eta} + \frac{dG^2}{4\eta} \log \left( 1 + \tfrac{1}{2}D^2 \mu^2 \big( \max_t \|x_t\|_2^2 \big) \frac{T}{d} \right) + \left( \eta + \frac{\mu}{4} - \frac{\mu}{2} \right) \sum_{t=1}^{T} (\widehat{y}_t - y_t^\star)^2 + \sum_{t=1}^{T} \frac{c_t^2}{\mu} \,,$$

after which replacing $\eta = \frac{\mu}{8}$ completes the proof. $\qquad\square$

# D  Details of Section 6 (Selective Sampling)

The following Theorem is a detailed version of Theorem 7

**Theorem 11.** *Fix an arbitrary sequence $\ell_1, \ldots, \ell_T$ of $\mu$-strongly convex differentiable losses. Then the predictions $\widehat{y}_t$ of Algorithm 1 run with inputs $M \geq \max_t |\ell_t'(\widehat{y}_t)|$, $\eta = \frac{\mu}{4}$, feedback $g_t = \frac{o_t}{q_{t-1}} \ell_t'(\widehat{y}_t)$, and $\kappa_t = q_t$ satisfy*

$$\mathbb{E}\left[\sum_{t=1}^{T}(\ell_t(\widehat{y}_t) - \ell_t(y_t^\star))\right] \leq \left(\frac{4M^2(\log(K)+1)}{\mu^{3/2}\beta}\right)^2 + \frac{4M^2(\log(K)+1)}{\mu},$$

*provided $\max_i \max_t |\widehat{y}_t - y_t(i)| \leq M$.*

*Proof.* First observe that $q_{t-1} \geq q_t$ and thus $\kappa_t = q_t$ is a valid choice, where we define $q_0 = q_1$. By equation (8) we have

$$\sum_{t=1}^{T} \gamma(\widehat{y}_t - y_t(i))\ell_t'(\widehat{y}_t)\frac{o_t}{q_{t-1}} \leq \frac{\log(K)}{q_T} + \gamma^2 \sum_{t=1}^{T} \kappa_{t-1}(\widehat{y}_t - y_t(i))^2 g_t^2$$

$$\leq \frac{\log(K)}{q_T} + \gamma^2 \sum_{t=1}^{T} \frac{o_t}{q_{t-1}}(\widehat{y}_t - y_t(i))^2 \ell_t'(\widehat{y}_t)^2$$

$$\leq \frac{\log(K)}{q_T} + \gamma^2 \sum_{t=1}^{T} \frac{o_t}{q_t}(\widehat{y}_t - y_t(i))^2 M^2,$$

where in the final inequality we used $q_t \leq q_{t-1}$ and $|\ell_t'(\widehat{y}_t)| \leq M$. After dividing both sides of the above inequality by $\gamma = \frac{\eta}{M^2}$ we find

$$\sum_{t=1}^{T}(\widehat{y}_t - y_t(i))\ell_t'(\widehat{y}_t)\frac{o_t}{q_{t-1}} \leq \frac{M^2\log(K)}{\eta q_T} + \eta \sum_{t=1}^{T} \frac{o_t}{q_t}(\widehat{y}_t - y_t(i))^2. \tag{17}$$

Following the analysis of the clipping trick by Cutkosky [2019], we have that

$$\sum_{t=1}^{T}(\widehat{y}_t - y_t(i))\ell_t'(\widehat{y}_t)\left(\frac{o_t}{q_t} - \frac{o_t}{q_{t-1}}\right) \leq M^2 \sum_{t=1}^{T}\left(\frac{1}{q_t} - \frac{1}{q_{t-1}}\right) \leq \frac{M^2}{q_T}, \tag{18}$$

where we used that the sum telescopes, that $q_t \leq q_{t-1}$, and that $q_0 = q_1$. Summing side by side (17) and (18) we find

$$\sum_{t=1}^{T}(\widehat{y}_t - y_t(i))\ell_t'(\widehat{y}_t)\frac{o_t}{q_t} \leq \frac{M^2(\log(K)+1)}{\eta q_T} + \eta \sum_{t=1}^{T} \frac{o_t}{q_t}(\widehat{y}_t - y_t(i))^2.$$

Since $\ell_t$ is $\mu$-strongly convex we have that

$$\mathbb{E}\left[\ell_t(\widehat{y}_t) - \ell_t(y_t(i))\right] \leq \mathbb{E}\left[(\widehat{y}_t - y_t(i))\ell_t'(\widehat{y}_t) - \frac{\mu}{2}(\widehat{y}_t - y_t(i))^2\right],$$

and therefore

$$\mathbb{E}\left[\sum_{t=1}^{T}(\ell_t(\widehat{y}_t) - \ell_t(y_t(i)))\right] \leq \mathbb{E}\left[\frac{M^2(\log(K)+1)}{\eta q_T}\right] + \left(\eta - \frac{\mu}{2}\right)\sum_{t=1}^{T}\mathbb{E}\left[(\widehat{y}_t - y_t(i))^2\right],$$

where we used that $\mathbb{E}_{t-1}[o_t] = q_t$. Now, by using $\min\{a,b\}^{-1} \leq a^{-1} + b^{-1}$ for $a, b > 0$ and Jensen's inequality we have that

$$\mathbb{E}\left[q_T^{-1}\right] = \mathbb{E}\left[\left(\min\left\{1, \beta \bigg/ \sqrt{\min_i \sum_{t=1}^{T}(\widehat{y}_t - y_t(i))^2}\right\}\right)^{-1}\right]$$

$$\leq 1 + \beta^{-1}\sqrt{\mathbb{E}\left[\min_i \sum_{t=1}^{T}(\widehat{y}_t - y_t(i))^2\right]},$$

and thus

$$\mathbb{E}\left[\sum_{t=1}^{T}(\ell_t(\widehat{y}_t) - \ell_t(y_t^\star))\right] \leq \frac{M^2(\log(K) + 1)}{\eta}$$

$$+ \beta^{-1}\sqrt{\mathbb{E}\left[\min_i \sum_{t=1}^{T}(\widehat{y}_t - y_t(i))^2\right]\frac{M^2(\log(K) + 1)}{\eta}} - \mathbb{E}\left[\sum_{t=1}^{T}\frac{\mu}{4}(\widehat{y}_t - y_t^\star)^2\right].$$

Using $ab \leq \frac{a^2\mu}{4} + \frac{b^2}{\mu}$ for $a, b > 0$ we continue

$$\mathbb{E}\left[\sum_{t=1}^{T}(\ell_t(\widehat{y}_t) - \ell_t(y_t^\star))\right] \leq \left(\frac{M^2(\log(K) + 1)}{\eta\beta\sqrt{\mu}}\right)^2 + \frac{M^2(\log(K) + 1)}{\eta}.$$

Using that $\eta^{-1} = \frac{4}{\mu}$ we arrive at the conclusion of the proof:

$$\mathbb{E}\left[\sum_{t=1}^{T}(\ell_t(\widehat{y}_t) - \ell_t(y_t^\star))\right] \leq \left(\frac{4M^2(\log(K) + 1)}{\mu^{3/2}\beta}\right)^2 + \frac{4M^2(\log(K) + 1)}{\mu}.$$

$\square$

### D.1 Selective Sampling for Online Regression

We now extend our selective sampling results to online regression, where we assume that the feature vector $\boldsymbol{x}_t$ is revealed to the learner before issuing a prediction. We run Algorithm 2 and, similarly to Section 6, we request the loss at round $t$ by drawing a Bernoulli variable $o_t$ of parameter

$$q_t = \min\left\{1, \beta \Big/ \min_{\boldsymbol{w}\in\mathcal{W}_t\cap\{\boldsymbol{w}:\|\boldsymbol{w}\|_2\leq D\}} \sum_{s=1}^{t}(\widehat{y}_s - \langle\boldsymbol{w}, \boldsymbol{x}_s\rangle)^2\right\}, \tag{19}$$

for some $D, \beta > 0$. This gives the following expected regret guarantee.

**Theorem 12.** *Fix an arbitrary sequence $\ell_1, \ldots, \ell_T$ of $\mu$-strongly convex differentiable losses. Then the predictions $\widehat{y}_t$ of Algorithm 2 run with inputs $\eta = \frac{\mu}{4}$, $\sigma = D^2$, $G \geq \max_t |\ell_t'(\widehat{y}_t)|$, $Z = \frac{1}{2}G$, feedback $g_t = \frac{o_t}{q_t}\ell_t'(\widehat{y}_t)$ and $\kappa_t = q_t$, satisfy*

$$\mathbb{E}\left[\sum_{t=1}^{T}(\ell_t(\widehat{y}_t) - \ell_t(y_t^\star))\right] \leq \left(\frac{2\zeta_T}{\mu^{3/2}\beta}\right)^2 + \frac{2\zeta_T}{\mu}.$$

*for any $\boldsymbol{x}_1, \ldots, \boldsymbol{x}_T \in \mathbb{R}^d$, and for any $\boldsymbol{u} \in \mathcal{W}_T \equiv \bigcap_{t=1}^{T}\{\boldsymbol{w} : |\langle\boldsymbol{w}, \boldsymbol{x}_t\rangle| \leq Z\}$ such that $\|\boldsymbol{u}\|_2 \leq D$ and $y_t^\star = \langle\boldsymbol{u}, \boldsymbol{x}_t\rangle$ for all $t \geq 1$, where*

$$\zeta_T = G^2 + dG^2\log\left(1 + D^2\max_t\|\boldsymbol{x}_t\|_2^2\left(\frac{T}{2d} + \frac{G^2T^2}{2d\beta^2}\right)\right).$$

*Proof.* Starting from (9) and replacing $g_t$ by $\frac{o_t}{q_t}\ell_t'(\widehat{y}_t)$ we find

$$\sum_{t=1}^{T}\frac{o_t}{q_t}\langle\boldsymbol{w}_t - \boldsymbol{u}, \boldsymbol{x}_t\ell_t'(\widehat{y}_t)\rangle$$

$$\leq \frac{\|\boldsymbol{u}\|_2^2}{2\sigma\gamma} + \frac{d}{2\kappa_T\gamma}\log\left(1 + 2\sigma\gamma^2\left(\max_t\|\boldsymbol{x}_t g_t\|_2^2\right)\frac{T}{d}\right) + \gamma\sum_{t=1}^{T}\kappa_t\frac{o_t}{q_t^2}(\langle\boldsymbol{w}_t - \boldsymbol{u}, \boldsymbol{x}_t\ell_t'(\widehat{y}_t)\rangle)^2$$

$$\leq \frac{G^2}{2\eta} + \frac{dG^2}{2q_T\eta}\log\left(1 + D^2q_t^{-2}\left(\max_t\|\boldsymbol{x}_t\|_2^2\right)\frac{T}{2d}\right) + \gamma\sum_{t=1}^{T}\frac{o_t}{q_t}(\langle\boldsymbol{w}_t - \boldsymbol{u}, \boldsymbol{x}_t\rangle)^2,$$

where in the second inequality we used that $\kappa_t = q_t$, $\|\boldsymbol{u}\|_2 \leq D$, $\sigma^2 = D^2$, $\gamma = \frac{\eta}{G^2}$, $\eta \leq \frac{1}{2}$ by assumption on $\mu$, and $|\ell'_t(\widehat{y}_t)| \leq G$. Now, using that $\langle \boldsymbol{w}_t - \boldsymbol{u}, \boldsymbol{x}_t \ell'_t(\widehat{y}_t)\rangle = (\widehat{y}_t - y_t^\star)\ell'_t(\widehat{y}_t)$ and by taking the expectation of both sides of the above and using that $\mathbb{E}_{t-1}[o_t] = q_t$ we find

$$\mathbb{E}\left[\sum_{t=1}^T (\widehat{y}_t - y_t^\star)\ell'_t(\widehat{y}_t)\right]$$

$$\leq \mathbb{E}\left[\frac{G^2}{2q_T\eta}\right] + \mathbb{E}\left[\frac{dG^2}{2\eta q_T}\log\left(1 + q_T^{-2}D^2\max_t \|\boldsymbol{x}_t\|_2^2 \frac{T}{2d}\right)\right] + \mathbb{E}\left[\sum_{t=1}^T \eta(\widehat{y}_t - y_t^\star)^2\right]$$

$$\leq \mathbb{E}\left[\frac{G^2}{2q_T\eta}\right] + \mathbb{E}\left[\frac{dG^2}{2\eta q_T}\log\left(1 + D^2\max_t \|\boldsymbol{x}_t\|_2^2 \left(\frac{T}{2d} + \frac{G^2T^2}{2d\beta^2}\right)\right)\right] + \mathbb{E}\left[\sum_{t=1}^T \eta(\widehat{y}_t - y_t^\star)^2\right]$$

$$= \mathbb{E}\left[\frac{\zeta_T}{2q_T\eta}\right] + \mathbb{E}\left[\sum_{t=1}^T \eta(\widehat{y}_t - y_t^\star)^2\right],$$

where in the final inequality we used that $q_T^{-1} \leq 1 + G\sqrt{T}/\beta$ and defined

$$\zeta_T = G^2 + dG^2\log\left(1 + D^2\max_t \|\boldsymbol{x}_t\|_2^2\left(\frac{T}{2d} + \frac{G^2T^2}{2d\beta^2}\right)\right).$$

Since $\ell_t$ is $\mu$-strongly convex we have that

$$\mathbb{E}[\ell_t(\widehat{y}_t) - \ell_t(y_t^\star)] \leq \mathbb{E}\left[(\widehat{y}_t - y_t^\star)\ell'_t(\widehat{y}_t) - \frac{\mu}{2}(\widehat{y}_t - y_t^\star)^2\right],$$

and therefore

$$\mathbb{E}\left[\sum_{t=1}^T (\ell_t(\widehat{y}_t) - \ell_t(y_t^\star))\right] \leq \mathbb{E}\left[\frac{\zeta_T}{2q_T\eta}\right] + \mathbb{E}\left[\sum_{t=1}^T \left(\eta - \frac{\mu}{2}\right)(\widehat{y}_t - y_t^\star)^2\right]$$

$$\leq \mathbb{E}\left[\frac{\zeta_T}{2q_T\eta}\right] - \mathbb{E}\left[\sum_{t=1}^T \frac{\mu}{4}(\widehat{y}_t - y_t^\star)^2\right],$$

where we used that $\eta = \frac{\mu}{4}$. Now, by using $\min\{a,b\}^{-1} \leq a^{-1} + b^{-1}$ for $a, b > 0$ and Jensen's inequality we have that

$$\mathbb{E}\left[q_T^{-1}\right] = \mathbb{E}\left[\left(\min\left\{1, \beta \Big/ \sqrt{\min_{\boldsymbol{w}\in\mathcal{W}_T\cap\{\boldsymbol{w}:\|\boldsymbol{w}\|_2\leq D\}} \sum_{t=1}^T (\widehat{y}_t - \langle\boldsymbol{w},\boldsymbol{x}_t\rangle)^2}\right\}\right)^{-1}\right]$$

$$\leq 1 + \beta^{-1}\sqrt{\mathbb{E}\left[\min_{\boldsymbol{w}\in\mathcal{W}_T\cap\{\boldsymbol{w}:\|\boldsymbol{w}\|_2\leq D\}} \sum_{t=1}^T (\widehat{y}_t - \langle\boldsymbol{w},\boldsymbol{x}_t\rangle)^2\right]},$$

and thus

$$\mathbb{E}\left[\sum_{t=1}^T (\ell_t(\widehat{y}_t) - \ell_t(y_t^\star))\right]$$

$$\leq \beta^{-1}\sqrt{\mathbb{E}\left[\min_{\boldsymbol{w}\in\mathcal{W}_T\cap\{\boldsymbol{w}:\|\boldsymbol{w}\|_2\leq D\}} \sum_{t=1}^T (\widehat{y}_t - \langle\boldsymbol{w},\boldsymbol{x}_t\rangle)^2\right]}\frac{\zeta_T}{2\eta} + \frac{\zeta_T}{2\eta} - \mathbb{E}\left[\sum_{t=1}^T \frac{\mu}{4}(\widehat{y}_t - y_t^\star)^2\right].$$

Using $ab \leq \frac{a^2\mu}{4} + \frac{b^2}{\mu}$ for $a, b > 0$ we continue

$$\mathbb{E}\left[\sum_{t=1}^T (\ell_t(\widehat{y}_t) - \ell_t(y_t^\star))\right] \leq \left(\frac{\zeta_T}{2\eta\beta\sqrt{\mu}}\right)^2 + \frac{\zeta_T}{2\eta}.$$

Using that $\eta^{-1} = \frac{4}{\mu}$ we arrive at the conclusion of the proof:

$$\mathbb{E}\left[\sum_{t=1}^T (\ell_t(\widehat{y}_t) - \ell_t(y_t^\star))\right] \leq \left(\frac{2\zeta_T}{\mu^{3/2}\beta}\right)^2 + \frac{2\zeta_T}{\mu}.$$

$\square$

---

**Algorithm 4:** AdaHedge with abstention [Van der Hoeven, 2020]

---

**Input** AdaHedge
**for** $t = 1, \ldots, T$ **do**
    Receive expert predictions $y_t(1), \ldots, y_t(K)$
    Obtain distribution $\boldsymbol{p}_t$ from AdaHedge
    Set $\widehat{y}_t = \sum_{i=1}^K p_t(i) y_t(i)$
    Set $\widetilde{y}_t = \text{sign}(\widehat{y}_t)$
    Set $b_t = 1 - |\widehat{y}_t|$
    Set sample $a_t$ from a Bernoulli distribution with parameter $1 - b_t$
    If $a_t = 1$, predict $\widetilde{y}_t$, otherwise abstain from prediction
    Receive $y_t$, send $\frac{1}{2}(1 - y_t(i)y_t)$ as the loss of the $i$-th expert to AdaHedge

---

# E  Exponentially Weighted Average

Here we provide a brief description of the Exponentially Weighted Average (EWA) algorithm [Vovk, 1990, Littlestone and Warmuth, 1994] on a discrete set of experts. EWA maintains a distribution $\boldsymbol{p}_t$ over the experts, where the mass on expert $i$ is given by

$$p_t(i) \propto \exp(-\eta \sum_{s=1}^{t-1} \ell_s(y_s(i))),$$

where $\eta > 0$ is the learning rate. A standard result is that the regret of EWA can be bounded as—see, for example, Van der Hoeven et al. [2018, Lemma 1]:

$$\mathcal{R}_T \leq \frac{\log(K)}{\eta} + \sum_{t=1}^T \left( \ell_t(\widehat{y}_t) + \frac{1}{\eta} \log(\mathbb{E}_{i \sim \boldsymbol{p}_t}\left[\exp(-\eta \ell_t(y_t(i)))\right]) \right) . \tag{20}$$

For $\alpha$-exp concave losses, which are losses for which $g_t(y) = \exp(-\alpha \ell_t(y))$ is concave, we can further bound (20) by choosing $\eta = \alpha$ and using Jensen's inequality:

$$\mathcal{R}_T \leq \frac{\log(K)}{\alpha} .$$

For $\mu$-strongly convex and $G$-Lipschitz losses we recover the optimal rate by using the fact that $\mu$-strongly convex losses are $\frac{\mu}{G^2}$-exp concave [Bubeck, 2011, proposition 1.2]:

$$\mathcal{R}_T \leq \frac{G^2 \log(K)}{\mu} .$$

# F  High Probability Regret Bounds for Online Prediction with Abstention

We consider the following generalization of the online learning with abstention setting due to Van der Hoeven [2020]. In each round $t = 1, \ldots, T$ the learner receives expert predictions $y_t(i) \in [-1, 1]$ and the learner can then either predict $\widetilde{y}_t \in [-1, 1]$ or abstain from prediction. If the learner predicts with $\widetilde{y}_t$ the learner suffers half the hinge loss $\ell_t(y) = \frac{1}{2}(1 - yy_t)$, where $y_t \in \{-1, 1\}$. If the learners abstains from prediction the learner suffers abstentions cost $\rho \in [0, \frac{1}{2})$. Let $a_t = 1$ if the learner predicts with $\widetilde{y}_t$ and let $a_t = 0$ if the learner abstains from prediction. The goal is to control the following definition of regret:

$$\sum_{t=1}^T a_t \ell_t(\widetilde{y}_t) + (1 - a_t)\rho - \sum_{t=1}^T \ell_t(y_t^\star) ,$$

where $y_t^\star = y_t(i^\star)$ and $i^\star = \text{argmin}_i \sum_{t=1}^T \ell_t(y_t(i))$. In the online prediction with abstention setting an analog of Lemma 1 can be derived. However, in the technical part we instead use Algorithm 2 by Van der Hoeven [2020]. Algorithm 2 by Van der Hoeven [2020] samples $a_t = 1$ with probability $1 - b_t$ and $a_t = 0$ with probability $b_t$, where $b_t = 1 - |\widehat{y}_t|$ and $\widehat{y}_t = \sum_{i=1}^K p_t(i) y_t(i)$. Distributions $\boldsymbol{p}_t$ come from AdaHedge [De Rooij et al., 2014] and if $a_t = 1$ we predict with $\widetilde{y}_t = \text{sign}(\widehat{y}_t)$. Algorithm 2 by Van der Hoeven [2020], or Algorithm 4 in this paper, has the following expected regret guarantee.

**Lemma 7.** *For any $\eta > 0$ Algorithm 4 guarantees*

$$\sum_{t=1}^{T}((1 - b_t)\ell_t(\widetilde{y}_t) + b_t\rho)$$

$$\leq \sum_{t=1}^{T} \ell_t(y_t^\star) + \frac{\log(K)}{\eta} + \frac{4}{3}\log(K) + 2 + \sum_{t=1}^{T} \tfrac{1}{2}\left(\eta - (1 - 2\rho)\right)\left(1 - |\widehat{y}_t|\right).$$

*Proof.* From [Van der Hoeven, 2020, Lemma 3] we have that for any $\eta > 0$

$$\sum_{t=1}^{T}((1 - b_t)\ell_t(\text{sign}(\widehat{y}_t)) + b_t\rho) \leq \sum_{t=1}^{T} \ell_t(y_t^\star) + \frac{4}{3}\log(K) + 2$$

$$+ \frac{\log(K)}{\eta} + \eta \sum_{t=1}^{T} \left( \mathbb{E}_{i \sim \boldsymbol{p}_t}\left[(\ell_t(\widehat{y}_t) - \ell_t(y_t(i)))^2\right] + ((1 - b_t)\ell_t(\text{sign}(\widehat{y}_t)) + b_t) - \ell_t(\widehat{y}_t) \right).$$

Now, by Van der Hoeven [2020, equation (16)] we have that

$$\mathbb{E}_{i \sim \boldsymbol{p}_t}\left[(\ell_t(\widehat{y}_t) - \ell_t(y_t(i)))^2\right] + ((1 - b_t)\ell_t(\text{sign}(\widehat{y}_t)) + b_t) - \ell_t(\widehat{y}_t)$$

$$\leq \rho(1 - |\widehat{y}_t|) + \eta\tfrac{1}{2}(1 - |\widehat{y}_t|) - \tfrac{1}{2}(1 - |\widehat{y}_t|),$$

and thus

$$\sum_{t=1}^{T}((1 - b_t)\ell_t(\text{sign}(\widehat{y}_t)) + b_t\rho)$$

$$\leq \sum_{t=1}^{T} \ell_t(y_t^\star) + \frac{\log(K)}{\eta} + \frac{4}{3}\log(K) + 2 + \sum_{t=1}^{T} \left(\eta\tfrac{1}{2} - \tfrac{1}{2}(1 - 2\rho)\right)\left(1 - |\widehat{y}_t|\right),$$

which completes the proof. $\qquad\square$

To see why Lemma 7 is the analog of Lemma 1 for the online prediction with abstention setting observe that by choosing $\eta < 1 - 2\rho$ we recover a bound akin to (3). In particular, by using Lemma 7, choosing $\eta = \frac{1}{2}(1 - 2\rho)$ we find

$$\sum_{t=1}^{T}((1 - b_t)\ell_t(\text{sign}(\widehat{y}_t)) + b_t\rho) \leq \sum_{t=1}^{T} \ell_t(y_t^\star) + \frac{2\log(K)}{(1 - 2\rho)} + \frac{4}{3}\log(K) + 2$$

$$- \frac{(1 - 2\rho)}{4}\sum_{t=1}^{T}(1 - |\widehat{y}_t|).$$

meaning we can exploit the negative $\frac{(1-2\rho)}{4}\sum_{t=1}^{T}(1 - |\widehat{y}_t|)$ in online prediction with abstention in a similar manner as we exploited the negative variance term in online learning with strongly convex losses. As an application of Lemma 7 we provide a high-probability bound for online prediction with abstention.

**Theorem 13.** *For $\delta \in (0, 1)$, with probability at least $1 - \delta$, Algorithm 4 guarantees*

$$\sum_{t=1}^{T}(a_t\ell_t(\text{sign}(\widehat{y}_t)) + (1 - a_t)\rho) \leq \frac{2\log(K)}{1 - 2\rho} + \frac{21\log(1/\delta)}{8(1 - 2\rho)} + \frac{4}{3}\log(K) + 2.$$

*Proof.* Let $r_t = a_t\ell_t(\text{sign}(\widehat{y}_t)) + (1 - a_t)\rho - ((1 - b_t)\ell_t(\text{sign}(\widehat{y}_t)) + b_t\rho)$. Since $\mathbb{E}[a_t\ell_t(\text{sign}(\widehat{y}_t)) + (1 - a_t)\rho] = ((1 - b_t)\ell_t(\text{sign}(\widehat{y}_t)) + b_t\rho)$ and $|r_t| \leq 1$, by Lemma 4, for $\delta \in (0, 1)$ and $\lambda \in [0, 1]$, with probability at least $1 - \delta$ we have that

$$r_t \leq \frac{\log(1/\delta)}{\lambda} + \lambda\frac{3}{4}\sum_{t=1}^{T}\mathbb{E}[r_t^2]$$

Now, let us study $\mathbb{E}[r_t^2] = \mathbb{E}[(a_t\ell_t(\mathrm{sign}(\widehat{y}_t)) + (1-a_t)\rho)^2] - ((1-b_t)\ell_t(\mathrm{sign}(\widehat{y}_t)) + b_t\rho)^2$. If $\mathrm{sign}(\widehat{y}_t) = y_t$ then

$$\mathbb{E}[r_t^2] = b_t\rho^2 - (b_t\rho)^2 \leq \frac{1}{4}b_t = \frac{1}{4}(1 - |\widehat{y}_t|).$$

If $\mathrm{sign}(\widehat{y}_t) \neq y_t$ then

$$\begin{aligned}
\mathbb{E}[r_t^2] =&(1-b_t) + b_t\rho^2 - (1 - b_t + b_t\rho)^2 \\
=&(1-b_t) + b_t\rho^2 - b_t^2\rho^2 - (1-b_t)^2 + (1-b_t)b_t\rho \\
=&|\widehat{y}_t| - |\widehat{y}_t|^2 + (1 - |\widehat{y}_t|)\rho^2 - (1 - |\widehat{y}_t|)^2\rho^2 + |\widehat{y}_t|(1 - |\widehat{y}_t|)\rho \\
\leq&|\widehat{y}_t|(1 - |\widehat{y}_t|) + (1 - |\widehat{y}_t|)\rho^2 \\
\leq&\frac{7}{4}(1 - |\widehat{y}_t|),
\end{aligned}$$

thus we may conclude that $\mathbb{E}[r_t^2] \leq \frac{7}{4}(1 - |\widehat{y}_t|)$. Combining the above with Lemma 7 we have that with probability at least $1 - \delta$

$$\sum_{t=1}^{T}(a_t\ell_t(\mathrm{sign}(\widehat{y}_t)) + (1-a_t)\rho) \leq \frac{\log(K)}{\eta} + \frac{\log(1/\delta)}{\lambda} + \frac{4}{3}\log(K) + 2$$

$$+ \sum_{t=1}^{T}(\eta\tfrac{1}{2} + \lambda\tfrac{21}{16} - \tfrac{1}{2}(1-2\rho)).$$

Since $\rho < 1/2$, $\eta = \frac{1}{2}(1-2\rho)$ and $\lambda = \frac{8}{21}(1-2\rho)$ are valid choices and we obtain

$$\sum_{t=1}^{T}(a_t\ell_t(\mathrm{sign}(\widehat{y}_t)) + (1-a_t)\rho) \leq \frac{2\log(K)}{1-2\rho} + \frac{21\log(1/\delta)}{8(1-2\rho)} + \frac{4}{3}\log(K) + 2,$$

which completes the proof. $\qquad\square$

## G Free Restarts

---
**Algorithm 5:** Restarting

---
**Input** $\eta > 0$, $M > 0$
**Initialize** $\tau_1 = 1$, $\nu = 1$, Algorithm 1 with inputs $M$ and $\eta$
**for** $t = 1, \ldots, T$ **do**
    Receive expert predictions $y_t(1), \ldots, y_t(K)$
    Send expert predictions to Algorithm 1 and receive $\widehat{y}_t$
    Predict $\widehat{y}_t$ and receive loss $\ell_t$
    Send $g_t = \ell_t'(\widehat{y}_t)$ and $\kappa_t = 1$ to Algorithm 1
    **if** $\min_i \dfrac{\mu}{4} \displaystyle\sum_{s=\tau_\nu}^{t}(\widehat{y}_s - y_s(i))^2 \geq \dfrac{4M^2\log(K)}{\mu}$ **then**
        Restart Algorithm 1 with inputs $M$ and $\eta$
        Set $\nu = \nu + 1$ and $\tau_{\nu+1} = t + 1$
Set $\tau_{\nu+1} = T + 1$

---

In this section, we introduce another way to exploit negative regret. The idea is the following. We keep track of $\min_i \sum_{s=1}^{t}(\widehat{y}_s - y_s(i))^2$, which is a lower bound on $\sum_{s=1}^{t}(\widehat{y}_s - y_s^\star)^2$, and as soon as $\frac{\mu}{4}\min_i \sum_{s=1}^{t}(\widehat{y}_s - y_s(i))^2 \geq \frac{4M^2\log(K)}{\mu}$, Lemma 1 with an appropriate $\eta$ ensures that $R_t \leq 0$. This implies that we may restart the algorithm for free, and compete with a new best expert from that point on. This approach leads to a simplified dynamic regret bound—see, for example, Zhang et al. [2018] or the references therein for a discussion of dynamic regret—in which the expert we are competing against may change in all rounds where the algorithm restarts. Our simplified dynamic regret bound is never larger than the standard regret bound. We denote by $\nu$ the number of restarts and by $\tau_\nu$ the first round of restart $\nu$. The algorithm can be found in Algorithm 5 and its regret guarantee can be found in Theorem 14 below.

**Theorem 14.** *Fix an arbitrary sequence $\ell_1, \ldots, \ell_T$ of $\mu$-strongly convex differentiable losses. Then the predictions $\widehat{y}_t$ of Algorithm 5 run with $g_t = \ell_t'(\widehat{y}_t)$, $\kappa_t = 1$, and inputs $\eta = \frac{\mu}{4}$ and $M$ guarantees*

$$\sum_{n=1}^{\nu} \max_i \sum_{t=\tau_n}^{\tau_{n+1}-1} \left( \ell_t(\widehat{y}_t) - \ell_t(y_t(i)) \right) \leq \frac{4M^2 \log(K)}{\mu} \,,$$

*provided that $\max_t \max_i |y_t(i)| \leq \frac{1}{2}M$ and $\max_t |\ell_t'(\widehat{y}_t)| \leq M$.*

*Proof.* First, observe that $\eta < \frac{1}{2}$ by assumption on $\mu$, making it a valid choice for Algorithm 1. For any $n < \nu$ we have that for any $i \in [K]$, by Lemma 2 and Lemma 1

$$\sum_{t=\tau_n}^{\tau_{n+1}-1} \left( \ell_t(\widehat{y}_t) - \ell_t(y_t(i)) \right) \leq \frac{4M^2 \log(K)}{\mu} - \frac{\mu}{4} \sum_{t=\tau_n}^{\tau_{n+1}-1} \left( \widehat{y}_t - y_t(i) \right)^2.$$

Since $n < \nu$ we must have that

$$\frac{4M^2 \log(K)}{\mu} \leq \min_j \frac{\mu}{4} \sum_{t=\tau_n}^{\tau_{n+1}-1} \left( \widehat{y}_t - y_t(j) \right)^2 \leq \frac{\mu}{4} \sum_{t=\tau_n}^{\tau_{n+1}-1} \left( \widehat{y}_t - y_t(i) \right)^2$$

and thus

$$\sum_{t=\tau_n}^{\tau_{n+1}-1} \left( \ell_t(\widehat{y}_t) - \ell_t(y_t(i)) \right) \leq 0 \,.$$

For $n = \nu$ we have that for any $i \in [K]$, by Lemma 2 and Lemma 1

$$\sum_{t=\tau_n}^{\tau_{n+1}-1} \left( \ell_t(\widehat{y}_t) - \ell_t(y_t(i)) \right) \leq \frac{4M^2 \log(K)}{\mu}$$

which completes the proof. $\qquad\square$