# OpenReview forum: "A Regret-Variance Trade-Off in Online Learning"
_NeurIPS.cc/2022/Conference — NeurIPS 2022 Accept_

### Official Review · Reviewer_JCyM · 2022-07-09

**Rating:** 7
**Confidence:** 4
**Soundness:** 4 excellent
**Presentation:** 2 fair
**Contribution:** 4 excellent

**Summary:**

This paper introduces a new proof technique applicable to many online learning problems where the loss function is strongly convex and bounded, and they show how this new analysis leads to an additional negative term in the regret bound. In different applications, they show how to exploit this negative term to obtain improved bounds for existing algorithms. In particular, for the online prediction with expert advice problem with a well-known $O(\log K)$ regret bound (where $K$ is the number of experts), they show how to obtain either a negative regret or an $O(\log K)$ bound for both variance (-variance is the new negative term in the bound) and regret. In addition, the authors show how to apply the same technique to online regression, statistical learning, online learning with corrupted feedback, selective sampling, and online prediction with abstention.

**Questions:**

- Could the results be generalized to exp-concave loss functions or is the strong convexity assumption necessary?
- Algorithm 1 introduced in the paper is a variation of the EWA algorithm and it is clear how the new idea obtains tighter bounds for these algorithms. Could the same idea be applied to the Multiplicate Weights algorithm and its variations (where $p_{t+1}(i)\propto p_t(i)(1-\kappa_t \tilde{\ell}_t(i))$)?
- The term ``mixability'' is used a couple of times on page 2 without a proper definition and explanation.
- There is a Lemma 1 on page 2 and there is another one on page 4.
- Using both $S$ and $\mathcal{S}$ in Algorithm 3 is a bit confusing and it's hard to distinguish these two from each other.


**Limitations:**

Yes, the limitations are stated.
No need for potential negative societal impacts discussion for this theoretical work.

**Strengths And Weaknesses:**

Strengths:
- The introduced proof technique is significant and novel and it has been motivated throughout the paper via several applications.
- While most of the technical details have been provided in the appendix, the authors have done a great job providing proof sketches and intuition to highlight how the new idea is applied in several applications and where the improvements come from.
- Looking into numerous learning problems in detail and showing how the new idea addresses many of the current research questions in each of these areas.

Weaknesses:
- The paper is a bit hard to follow because of the amount of information that is provided in the text. The authors have tried to highlight all of the applications that benefit from this new idea, however, given the page limit, the discussion regarding each application is very short, and readers with limited exposure to each of these research areas may find it hard to follow the discussions and appreciate the new contributions.
- Some numerical experiments (particularly for the model selection aggregation problem and Algorithm 3) would have been very helpful to verify the theoretical findings in practice.

---

> ### Author Response · Authors · 2022-08-01
> **Response to reviewer JCyM**
>
> Thank you for your careful reading of the paper.
>
> We do not know if the strong convexity assumption is necessary, but it does play a critical role in our results. The key challenge for relaxing the curvature assumption lies in deriving inequalities similar to equation (1). Vijaykumar (2021) derives related inequalities for exp-concave and self-concordant functions, but at the moment we do not know how to exploit these inequalities with existing online learning algorithms (the algorithms of  Vijaykumar can only be applied in the batch setting).
>
> An algorithm similar to what you suggest is analyzed in Koolen and Van Erven (2015): if we use $p_{t+1}(i) \propto p_t(i)(1 + \gamma (y_t(i) - \hat{y}_t)g_t)$ we can obtain linearized regret bounds of the form (2).
>
> Thank you for your remaining remarks. We will address them in the next version of the paper.
>
> References:
>
> Vijaykumar, S. (2021). Localization, convexity, and star aggregation. Advances in Neural Information Processing Systems, 34, 4570-4581.
>
> Koolen, W. M., \& Van Erven, T. (2015). Second-order quantile methods for experts and combinatorial games. In Conference on Learning Theory (pp. 1155-1175). PMLR.

---

> > ### Comment · Reviewer_JCyM · 2022-08-08
> > **Re: Response**
> >
> > Thanks for your detailed response, I don't have any further questions.

---

### Official Review · Reviewer_9Zba · 2022-07-20

**Rating:** 4
**Confidence:** 3
**Soundness:** 3 good
**Presentation:** 1 poor
**Contribution:** 2 fair

**Summary:**

For the prediction with expert advice problem, the paper considers a modification of the Exponentially Weighted Average algorithm and shows that when variance is large, it is possible to achieve negative regret, i.e., outperform the best expert, otherwise, variance is small (bounded by O(log K), and they fall back to the well-known O(log K) regret bound.

Further, they use this concept (with the algorithm and its bound) in several other settings and prove regret bounds, essentially trying to make algorithms make use of large observed variance in the loss.

**Questions:**

1.	Can it explicitly be stated upfront why strongly convex property of losses is required and where it is used?
2.	In my review (and score), based on my complete reading and understanding, I have assumed Sections 3 and 4 are the crux of the work, and Sections 5, 6, and 7 do not form the main contribution of the work. Is that fair? If not, please help me understand.
3.	In Appendix, equation 7, how does the KL divergence term taken from the literature become $\ln K$ here? Please elaborate.
4.	In Appendix, line 508, What choice of $\gamma$ and how does the previous inequality hold? Did I miss something here?

In the Appendix, I only checked the proof of Lemma 1, and did not read the other proofs.

Minor suggestions:

1.	Line 21 – Explicitly mention what M is.

2.	Line 37 - How does assuming $\mu \leq 2$ qualify as "without loss of generality"?

3.	Line 56 - “Informal lemma”?

4.	"Our Algorithms" section doesn’t contain all algorithms. Consider re-arranging them to accommodate Algo 3, or perhaps say “Core Algorithms/Results”?

5.	Lines 201-204 and Lines 226-230 have commonalities. Consider defining it once and referring from both places.

6.	$\cal{F}$ is defined as a family/set/dictionary in different places like Line 201, 227. Maintain uniformity.

7.	Line 201, 227 – Further elaborate/define an instance space $\Chi$.

8.	Lines 203, 229 – Improve readability. Move $f: \Chi \Rightarrow \Real$ two lines up to “real-valued functions defined on..” part.

9.	Lines 204, 229 – Expectation term is missing square brackets?

10.	Line 234 – Slightly elaborate on why the term is called `optimal rate of aggregation’. I didn’t follow.

11.	Appendix, Line 510 – “we write”.



**Limitations:**

The authors have given adequate references to papers where the original algorithms are taken from, to papers that have considered almost similar settings, to papers whose results are recovered by the current work.

The authors have been upfront about the limitations of the work.


**Strengths And Weaknesses:**

Strengths:

1.	Centered around a common theme of utilizing large variance, the paper shows the applicability of this result in several different settings to get auxiliary results.

Weakness:

1.	The writing and clarity can be improved. For instance, almost all of the technical stuff is in the Appendix. While I realize the constraints of the page limit, most of the main paper content was overloaded. I suggest showing Lemma 1 and its proof in greater detail in the main paper and moving some of the auxiliary results entirely into the Appendix.

2.	All the existing literature work mentioned considers slightly different settings. Given there is no apples-to-apples comparison of the current work with literature, it becomes difficult to assess the quality of the work in the absence of showing lower bounds or showing how close to optimal these algorithms are.

---

> ### Author Response · Authors · 2022-08-01
> **Response to reviewer 9Zba**
>
>
> Thank you for your careful reading of the paper.
>
> We would first like to address the weaknesses you mention in your review before we move on to answering your questions.
>
> While a lot of technical details are indeed in the appendix, our goal in the main body was to convey the message that the negative quadratic term obtained from strong convexity (in particular equation (1)) can be used not only to obtain logarithmic regret bounds, but also in several applications. The main technical idea is thus really to exploit this negative quadratic term to compensate for positive quadratic terms that are found while analyzing these applications.
>
> Due to a typo, two distinct results in the main body of the paper are both called Lemma 1: one in the introduction and one in Section 3. Given the space constraints, we do not think that the proof of Lemma 1 in Section 3 should appear in the main body, as it can also be found in other works, e.g., in Koolen and Van Erven (2015), and does not convey our main ideas.
> The proof of Lemma 1 in the introduction is a combination of equations (1) and (2)
> $$
> R_T \leq \tilde{R}_T - \frac{\mu}{2}  \sum_t (\hat{y}_t - y^\star_t)^2 \leq \frac{C_T}{\eta} + B_T - \Big(\frac{\mu}{2} - \eta\Big)\sum_t(\hat{y}_t - y^\star_t)^2,
> $$
> where the first inequality is due to equation (1) and the second inequality is due to equation (2). We will consider adding this proof to the main body as it is the central inequality of our paper.
>
> As for the comparison with the literature, the statistical learning setting of Section 4 matches the literature exactly. We tried to explain this in the discussion of the related work in that section. We will clarify that because of that, the lower bounds developed in other works also apply to our results. The selective sampling setting of Section 6 is exactly the same setting considered in prior work, up to the assumptions on the curvature of the loss. The abstention setting of Section 7 also matches the literature exactly. To the best of our knowledge, instead, the version of corrupted feedback that we consider does not appear in the literature.
>
> We will now try to answer your questions.
>
> The strong convexity of the loss is required to obtain the negative quadratic term we exploit throughout the paper. It is used in the proof of Lemma 1, specifically in the step where we apply equation (1).
>
> All the sections in the main body of the paper provide key contributions to our work. We introduce the idea of exploiting the negative term that appears in Lemma 1 in the introduction. In Section 3 we describe the algorithms that we use. In sections 4, 5, and 6 we indeed show that the negative quadratic term can be exploited and provide meaningful contributions to the settings considered in these settings. In section 7 we provide an extension of our ideas to the prediction with abstention setting, which answers an open question and shows the power of our ideas.
>
> For distributions over discrete spaces with cardinality $K$, the KL divergence between any distribution and the uniform distribution is bounded by $\ln(K)$. To see why, let us denote by $q$ the uniform distribution. We have that
> $$
> KL(p||q) = \sum_{i = 1}^K p(i) \ln\big( K{p(i)} \big) = \sum_{i = 1}^K p(i) \ln {p(i)} + \ln K \leq  \ln(K),
> $$
> where the inequality is due to the fact that $\ln(x) \leq 0$ for $x \in (0, 1]$.
>
> The choice of $\gamma$ is given in the algorithm and is $\gamma = \eta/M^2$. We are not entirely certain what you mean by the previous inequality, so we will explain both inequalities in equation 507. The first inequality is the so-called prod bound, which can be found in Lemma 2.4 by Cesa-Bianchi and Lugosi (2006), we will add this reference. The second inequality is because of our assumptions on $\hat{y}_t, y^\star_t$, and $g_t$:
> $$
> |\gamma (\hat{y}_t - y^\star_t)g_t| \leq \gamma M^2 = \eta \leq \frac{1}{2},
> $$
> where the final inequality is due to the fact that $\eta \in (0, \frac{1}{2}]$.
>
> Thank you for your minor suggestions. Should you have any further questions we are happy to address them if possible.
>
> References:
>
> Koolen, W. M., \& Van Erven, T. (2015). Second-order quantile methods for experts and combinatorial games. In Conference on Learning Theory (pp. 1155-1175). PMLR.
>
> Cesa-Bianchi, N., \& Lugosi, G. (2006). Prediction, learning, and games. Cambridge university press.

---

### Official Review · Reviewer_ToYU · 2022-07-20

**Rating:** 7
**Confidence:** 4
**Soundness:** 4 excellent
**Presentation:** 2 fair
**Contribution:** 3 good

**Summary:**

The authors consider the problems of prediction with expert advice and online linear regression with bounded, differentiable and strongly convex losses. They analyzed variants of the Squint algorithm in the expert advice problem and MetaGrad algorithm in the online linear regression problem. Using curvature of the strongly-convex loss functions and properly tuning the learning rate depending on the strong-convexity parameter, they make appear a negative term in the respective regret bounds of both the algorithm, whose magnitude is proportional the squared difference of learner’s predictions and the best (expert's) predictions. The authors call this term as the "variance" and present trade-offs between regret and variance in applications like online-to-batch conversion, online learning with corrupted loss and with abstentions.

In $K$-experts problem, the authors show that Squint algorithm enjoys negative regret if the "variance" term is large. Else, they prove that both regret and variance are upper bounded by $\log K$. They apply this result to show that early stopping can be achieved in standard online-to-batch conversion scheme when variance is large. If variance is bounded, then the authors show optimal high-probability excess risk bound can be achieved by online-to-batch conversion scheme.

In online regression problem, the authors show that MetGrad enjoys negative regret if the corresponding variance term is large, and apply this result to obtain optimal high probability excess risk bound using online-to-batch-conversion scheme.

As corollaries of above results, the authors show how the negative variance term can mitigate the effect of adversarial corruptions and abstentions in corresponding regret bounds.

**Questions:**

I would like the authors to comment whether the curvature assumption can be relaxed or not, and whether the results can be extended to tackle, say, heavy-tailed corruptions.

**Strengths And Weaknesses:**

Strengths:

1. The authors give a nice trick to intelligently combine the curvature of the loss function with a quadratic term that appear in the standard regret analysis of online learning algorithms, which yields some improvements over existing regret bounds in the literature.

2. This intelligent trick is then applied to derive first high-probability optimal excess risk bound in model selection aggregation problem, which is a novel contribution of this paper.

3. The trick has useful applications in interesting online learning problems with corrupted feedback and with abstentions.

Weaknesses:

1. The main results of this paper heavily depend on strong convexity of the loss functions making practical applicability for the results limited.

2. The results also assume boundedness of the gradients, which is fine. However, when applied to the corrupted feedback setting, this assumptions restricts the corruptions to be bounded too. In many practical scenarios, however, corruptions can be unbounded (with some non-zero probability). The current frame work can't capture this properly.

---

> ### Author Response · Authors · 2022-08-01
> **Response to reviewer ToYU**
>
> Thank you for your careful reading of the paper.
>
> Thank you for your questions. At the moment we do not know whether the curvature assumptions can be weakened. The key challenge for relaxing the curvature assumption lies in deriving similar inequalities as in equation~(1). Vijaykumar (2021) derives related inequalities for exp-concave and self-concordant functions, but at the moment we do not know how to exploit these inequalities with existing online learning algorithms {(the algorithms of  Vijaykumar can only be applied in the batch setting)}.
>
> Thank you for your question on the corrupted feedback setting. While unbounded corruptions are indeed a problem, there is a line of work in online learning that deals with unbounded feedback (see Jun and Orabona (2019) and subsequent works) under different stochastic assumptions on the feedback. The algorithms employed in these works are similar to Squint and Metagrad, so there is hope that the techniques developed in these works transfer to our setting. However, we leave unbounded corruption as a future research question.
>
> References:
>
> Jun, K. S., \& Orabona, F. (2019). Parameter-free online convex optimization with sub-exponential noise. In Conference on Learning Theory (pp. 1802-1823). PMLR.
>
> Vijaykumar, S. (2021). Localization, convexity, and star aggregation. Advances in Neural Information Processing Systems, 34, 4570-4581.

---

### Meta-Review · Area_Chair_sNQQ · 2022-09-02

**Recommendation:** Accept
**Confidence:** Certain

**Metareview:**

The paper introduces the valuable idea of exploiting strong convexity of losses in online learning, together with variance-based regret bounds for contemporary algorithms like Squint and Metagrad, to introduce negative terms in cumulative regret bounds and make the algorithms useful in many applications such as early stopping in online-to-batch conversion and other settings.

A dominant concern from the reviewers' side was about the amount of (technical) material packed into the paper, which was alleviated by the detailed author response. As a result, the reviewers largely agree that the paper deserves to be accepted -- an opinion which I share.

PS. I request the author(s) to please resolve the incomplete [TODO]s in the paper checklist appropriately for the final version.

**Award:**

No

---

### Decision · Program_Chairs · 2022-09-14

Accept